# Rényi Differential Privacy Mechanisms for Posterior Sampling

**Joseph Geumlek**
University of California, San Diego
jgeumlek@cs.ucsd.edu

**Shuang Song**
University of California, San Diego
shs037@eng.ucsd.edu

**Kamalika Chaudhuri**
University of California, San Diego
kamalika@cs.ucsd.edu

## Abstract

With the newly proposed privacy definition of Rényi Differential Privacy (RDP) in [14], we re-examine the inherent privacy of releasing a single sample from a posterior distribution. We exploit the impact of the prior distribution in mitigating the influence of individual data points. In particular, we focus on sampling from an exponential family and specific generalized linear models, such as logistic regression. We propose novel RDP mechanisms as well as offering a new RDP analysis for an existing method in order to add value to the RDP framework. Each method is capable of achieving arbitrary RDP privacy guarantees, and we offer experimental results of their efficacy.

## 1  Introduction

As data analysis continues to expand and permeate ever more facets of life, the concerns over the privacy of one's data grow too. Many results have arrived in recent years to tackle the inherent conflict of extracting usable knowledge from a data set without over-extracting or leaking the private data of individuals. Before one can strike a balance between these competing goals, one needs a framework by which to quantify what it means to preserve an individual's privacy.

Since 2006, Differential Privacy (DP) has reigned as the privacy framework of choice [6]. It quantifies privacy by measuring how indistinguishability of the mechanism output across whether or not any one individual is in or out of the data set. This gave not just privacy semantics, but also robust mathematical guarantees. However, the requirements have been cumbersome for utility, leading to many proposed relaxations. One common relaxation is approximate DP, which allows arbitrarily bad events to occur with probability at most $\delta$. A more recent relaxation is Rényi Differential Privacy (RDP) proposed in [14], which uses the measure of Rényi divergences to smoothly vary between bounding the average and maximum privacy loss. However, RDP has very few mechanisms compared to the more established approximate DP. We expand the RDP repertoire with novel mechanisms inspired by Rényi divergences, as well as re-analyzing an existing method in this new light.

Inherent to DP and RDP is that there must be some uncertainty in the mechanism; they cannot be deterministic. Many privacy methods have been motivated by exploiting pre-existing sources of randomness in machine learning algorithms. One promising area has been Bayesian data analysis, which focuses on maintaining and tracking the uncertainty within probabilistic models. Posterior sampling is prevalent in many Bayesian methods, serving to introduce randomness that matches the currently held uncertainty.

We analyze the privacy arising from posterior sampling as applied to two domains: sampling from exponential families and Bayesian logistic regression. Along with these analyses, we offer tunable mechanisms that can achieve stronger privacy guarantees than directly sampling from the posterior. These mechanisms work via controlling the relative strength of the prior in determining the posterior, building off the common intuition that concentrated prior distributions can prevent overfitting in Bayesian data analysis. We experimentally validate our new methods on synthetic and real data.

## 2   Background

**Privacy Model.**  We say two data sets $\mathbf{X}$ and $\mathbf{X}'$ are *neighboring* if they differ in the private record of a single *individual* or person. We use $n$ to refer to the number of records in the data set.

**Definition 1.** *Differential Privacy (DP) [6]. A randomized mechanism $\mathcal{A}(\mathbf{X})$ is said to be $(\epsilon, \delta)$-differentially private if for any subset $U$ of the output range of $\mathcal{A}$ and any neighboring data sets $\mathbf{X}$ and $\mathbf{X}'$, we have $p(\mathcal{A}(\mathbf{X}) \in U) \leq exp\,(\epsilon)\,p(\mathcal{A}(\mathbf{X}') \in U) + \delta$.*

DP is concerned with the difference the participation of an individual might have on the output distribution of the mechanism. When $\delta > 0$, it is known as approximate DP while the $\delta = 0$ case is known as pure DP. The requirements for DP can be phrased in terms of a privacy loss variable, a random variable that captures the effective privacy loss of the mechanism output.

**Definition 2.** *Privacy Loss Variable [2]. We can define a random variable $Z$ that measures the privacy loss of a given output of a mechanism across two neighboring data sets $\mathbf{X}$ and $\mathbf{X}'$.*

$$Z = \log \frac{p(\mathcal{A}(\mathbf{X}) = o)}{p(\mathcal{A}(\mathbf{X}') = o)}\bigg|_{o \sim \mathcal{A}(\mathbf{X})} \tag{1}$$

$(\epsilon, \delta)$-DP is the requirement that for any two neighboring data sets $Z \leq \epsilon$ with probability at least $1 - \delta$. The exact nature of the trade-off and semantics between $\epsilon$ and $\delta$ is subtle, and choosing them appropriately is difficult. For example, setting $\delta = 1/n$ permits $(\epsilon, \delta)$-DP mechanisms that always violate the privacy of a random individual [12]. However, there are other ways to specify that a random variable is mostly small. One such way is to bound the Rényi divergence of $\mathcal{A}(\mathbf{X})$ and $\mathcal{A}(\mathbf{X}')$.

**Definition 3.** *Rényi Divergence [2]. The Rényi divergence of order $\lambda$ between the two distributions $P$ and $Q$ is defined as*

$$D_\lambda(P||Q) = \frac{1}{\lambda - 1} \log \int P(o)^\lambda Q(o)^{1-\lambda} do. \tag{2}$$

As $\lambda \to \infty$, Rényi divergence becomes the *max divergence*; moreover, setting $P = \mathcal{A}(\mathbf{X})$ and $Q = \mathcal{A}(\mathbf{X}')$ ensures that $D_\lambda(P||Q) = \frac{1}{\lambda-1} \log \mathbb{E}_Z[e^{(\lambda-1)Z}]$, where $Z$ is the privacy loss variable. Thus, a bound on the Rényi divergence over all orders $\lambda \in (0, \infty)$ is equivalent to $(\epsilon, 0)$-DP, and as $\lambda \to 1$, this approaches the expected value of $Z$ equal to $KL(\mathcal{A}(\mathbf{X})||\mathcal{A}(\mathbf{X}'))$. This leads us to Rényi Differential Privacy, a flexible privacy notion that covers this intermediate behavior.

**Definition 4.** *Rényi Differential Privacy (RDP) [14]. A randomized mechanism $\mathcal{A}(\mathbf{X})$ is said to be $(\lambda, \epsilon)$-Rényi differentially private if for any neighboring data sets $\mathbf{X}$ and $\mathbf{X}'$ we have $D_\lambda(\mathcal{A}(\mathbf{X})||\mathcal{A}(\mathbf{X}')) \leq \epsilon$.*

The choice of $\lambda$ in RDP is used to tune how much concern is placed on unlikely large values of $Z$ versus the average value of $Z$. One can consider a mechanism's privacy as being quantified by the entire curve of $\epsilon$ values associated with each order $\lambda$, but the results of [14] show that almost identical results can be achieved when this curve is known at only a finite collection of possible $\lambda$ values.

**Posterior Sampling.**  In Bayesian inference, we have a model class $\Theta$, and are given observations $x_1, \ldots, x_n$ assumed to be drawn from a $\theta \in \Theta$. Our goal is to maintain our beliefs about $\theta$ given the observational data in the form of the posterior distribution $p(\theta|x_1, \ldots, x_n)$. This is often done in the form of drawing samples from the posterior.

Our goal in this paper is to develop privacy preserving mechanisms for two popular and simple posterior sampling methods. The first is sampling from the exponential family posterior, which we address in Section 3; the second is sampling from posteriors induced by a subset of Generalized Linear Models, which we address in Section 4.

**Related Work.** Differential privacy has emerged as the gold standard for privacy in a number of data analysis applications – see [8, 15] for surveys. Since enforcing pure DP sometimes requires the addition of high noise, a number of relaxations have been proposed in the literature. The most popular relaxation is approximate DP [6], and a number of uniquely approximate DP mechanisms have been designed by [7, 16, 3, 1] among others. However, while this relaxation has some nice properties, recent work [14, 12] has argued that it can also lead privacy pitfalls in some cases. Approximate differential privacy is also related to, but is weaker than, the closely related $\delta$-probabilistic privacy [11] and $(1, \epsilon, \delta)$-indistinguishability [4].

Our privacy definition of choice is Rényi differential privacy [14], which is motivated by two recent relaxations – concentrated DP [9] and z-CDP [2]. Concentrated DP has two parameters, $\mu$ and $\tau$, controlling the mean and concentration of the privacy loss variable. Given a privacy parameter $\alpha$, z-CDP essentially requires $(\lambda, \alpha\lambda)$-RDP for all $\lambda$. While [2, 9, 14] establish tighter bounds on the privacy of existing differentially private and approximate DP mechanisms, we provide mechanisms based on posterior sampling from exponential families that are uniquely RDP. RDP is also a generalization of the notion of KL-privacy [19], which has been shown to be related to generalization in machine learning.

There has also been some recent work on privacy properties of Bayesian posterior sampling; however most of the work has focused on establishing pure or approximate DP. [5] establishes conditions under which some popular Bayesian posterior sampling procedures directly satisfy pure or approximate DP. [18] provides a pure DP way to sample from a posterior that satisfies certain mild conditions by raising the temperature. [10, 20] provide a simple statistically efficient algorithm for sampling from exponential family posteriors. [13] shows that directly sampling from the posterior of certain GLMs, such as logistic regression, with the right parameters provides approximate differential privacy. While our work draws inspiration from all [5, 18, 13], the main difference between their and our work is that we provide RDP guarantees.

## 3 RDP Mechanisms based on Exponential Family Posterior Sampling

In this section, we analyze the Rényi divergences between distributions from the same exponential family, which will lead to our RDP mechanisms for sampling from exponential family posteriors.

An exponential family is a family of probability distributions over $x \in \mathcal{X}$ indexed by the parameter $\theta \in \Theta \subseteq \mathbb{R}^d$ that can be written in this canonical form for some choice of functions $h : \mathcal{X} \to \mathbb{R}$, $S : \mathcal{X} \to \mathbb{R}^d$, and $A : \Theta \to \mathbb{R}$:

$$p(x_1, \ldots, x_n | \theta) = \left( \prod_{i=1}^n h(x_i) \right) \exp \left( (\sum_{i=1}^n S(x_i)) \cdot \theta - n \cdot A(\theta) \right) . \tag{3}$$

Of particular importance is $S$, the sufficient statistics function, and $A$, the log-partition function of this family. Our analysis will be restricted to the families that satisfy the following three properties.

**Definition 5.** *The natural parameterization of an exponential family is the one that indexes the distributions of the family by the vector $\theta$ that appears in the inner product of equation* (3).

**Definition 6.** *An exponential family is minimal if the coordinates of the function $S$ are not linearly dependent for all $x \in \mathcal{X}$.*

**Definition 7.** *For any $\Delta \in \mathbb{R}$, an exponential family is $\Delta$-bounded if* $\Delta \geq \sup_{x,y \in \mathcal{X}} ||S(x) - S(y)||$.

*This constraint can be relaxed with some caveats explored in the appendix.*

A minimal exponential family will always have a minimal conjugate prior family. This conjugate prior family is also an exponential family, and it satisfies the property that the posterior distribution formed after observing data is also within the same family. It has the following form:

$$p(\theta | \eta) = \exp \left( T(\theta) \cdot \eta - C(\eta) \right). \tag{4}$$

The sufficient statistics of $\theta$ can be written as $T(\theta) = (\theta, -A(\theta))$ and $p(\theta | \eta_0, x_1, \ldots, x_n) = p(\theta | \eta')$ where $\eta' = \eta_0 + \sum_{i=1}^n (S(x_i), 1)$.

**Beta-Bernoulli System.** A specific example of an exponential family that we will be interested in is the Beta-Bernoulli system, where an individual's data is a single i.i.d. bit modeled as a Bernoulli variable with parameter $\rho$, along with a Beta conjugate prior. $p(x|\rho) = \rho^x(1-\rho)^{1-x}$.

The Bernoulli distribution can be written in the form of equation (3) by letting $h(x) = 1$, $S(x) = x$, $\theta = \log(\frac{\rho}{1-\rho})$, and $A(\theta) = \log(1 + \exp(\theta)) = -\log(1-\rho)$. The Beta distribution with the usual parameters $\alpha_0, \beta_0$ will be parameterized by $\eta_0 = (\eta_0^{(1)}, \eta_0^{(2)}) = (\alpha_0, \alpha_0 + \beta_0)$ in accordance equation (4). This system satisfies the properties we require, as this natural parameterization is minimal and $\Delta$-bounded for $\Delta = 1$. In this system, $C(\eta) = \Gamma(\eta^{(1)}) + \Gamma(\eta^{(2)} - \eta^{(1)}) - \Gamma(\eta^{(2)})$.

**Closed Form Rényi Divergence.** The Rényi divergences of two distributions within the same family can be written in terms of the log-partition function.

$$D_\lambda(P\|Q) = \frac{1}{\lambda - 1} \log\left(\int_\Theta P(\theta)^\lambda Q(\theta)^{1-\lambda} d\theta\right) = \frac{C(\lambda\eta_P + (1-\lambda)\eta_Q) - \lambda C(\eta_P)}{\lambda - 1} + C(\eta_Q).$$

(5)

To help analyze the implication of equation (5) for Rényi Differential Privacy, we define some sets of prior/posterior parameters $\eta$ that arise in our analysis.

**Definition 8.** *Normalizable Set E. We say a posterior parameter $\eta$ is normalizable if $C(\eta) = \log \int_\Theta exp(T(\theta) \cdot \eta)) d\theta$ is finite. Let E contain all normalizable $\eta$ for the conjugate prior family.*

**Definition 9.** *Let $pset(\eta_0, n)$ be the convex hull of all parameters $\eta$ of the form $\eta_0 + n(S(x), 1)$ for $x \in \mathcal{X}$. When $n$ is an integer this represents the hull of possible posterior parameters after observing $n$ data points starting with the prior $\eta_0$.*

**Definition 10.** *Let $Diff$ be the difference set for the family, where $Diff$ is the convex hull of all vectors of the form $(S(x) - S(y), 0)$ for $x, y \in \mathcal{X}$.*

**Definition 11.** *Two posterior parameters $\eta_1$ and $\eta_2$ are neighboring iff $\eta_1 - \eta_2 \in Diff$. They are r-neighboring iff $\eta_1 - \eta_2 \in r \cdot Diff$.*

### 3.1 Mechanisms and Privacy Guarantees

We begin with our simplest mechanism, Direct Sampling, which samples according to the true posterior. This mechanism is presented as Algorithm 1.

---
**Algorithm 1** Direct Posterior
---
**Require:** $\eta_0, \{x_1, \ldots, x_n\}$
  1: Sample $\theta \sim p(\theta|\eta')$ where $\eta' = \eta_0 + \sum_{i=1}^n (S(x_i), 1)$
---

Even though Algorithm 1 is generally not differentially private [5], Theorem 12 suggests that it offers RDP for $\Delta$-bounded exponential families and certain orders $\lambda$.

**Theorem 12.** *For a $\Delta$-bounded minimal exponential family of distributions $p(x|\theta)$ with continuous log-partition function $A(\theta)$, there exists $\lambda^* \in (1, \infty]$ such Algorithm 1 achieves $(\lambda, \epsilon(\eta_0, n, \lambda))$-RDP for $\lambda < \lambda^*$.*

*$\lambda^*$ is the supremum over all $\lambda$ such that all $\eta$ in the set $\eta_0 + (\lambda - 1)Diff$ are normalizable.*

**Corollary 1.** *For the Beta-Bernoulli system with a prior $Beta(\alpha_0, \beta_0)$, Algorithm 1 achieves $(\lambda, \epsilon)$-RDP iff $\lambda > 1$ and $\lambda < 1 + min(\alpha_0, \beta_0)$.*

Notice the implication of Corollary 1: for any $\eta_0$ and $n > 0$, there exists finite $\lambda$ such that direct posterior sampling does not guarantee $(\lambda, \epsilon)$-RDP for any finite $\epsilon$. This also prevents $(\epsilon, 0)$-DP as an achievable goal. Algorithm 1 is inflexible; it offers us no way to change the privacy guarantee.

This motivates us to propose two different modifications to Algorithm 1 that are capable of achieving arbitrary privacy parameters. Algorithm 2 modifies the contribution of the data **X** to the posterior by introducing a coefficient $r$, while Algorithm 3 modifies the contribution of the prior $\eta_0$ by introducing a coefficient $m$. These simple ideas have shown up before in variations: [18] introduces a temperature

---

**Algorithm 2** Diffused Posterior

---

**Require:** $\eta_0, \{x_1, \ldots, x_n\}, \epsilon, \lambda$
  1: Find $r \in (0, 1]$ such that $\forall r$-neighboring $\eta_P, \eta_Q \in pset(\eta_0, rn), D_\lambda(p(\theta|\eta_P)||p(\theta|\eta_Q)) \leq \epsilon$
  2: Sample $\theta \sim p(\theta|\eta')$ where $\eta' = \eta_0 + r \sum_{i=1}^n (S(x_i), 1)$

---

scaling that acts similarly to $r$, while [13, 5] analyze concentration constraints for prior distributions much like our coefficient $m$.

**Theorem 13.** *For any $\Delta$-bounded minimal exponential family with prior $\eta_0$ in the interior of E, any $\lambda > 1$, and any $\epsilon > 0$, there exists $r^* \in (0, 1]$ such that using $r \in (0, r^*]$ in Algorithm 2 will achieve $(\lambda, \epsilon)$-RDP.*

---

**Algorithm 3** Concentrated Posterior

---

**Require:** $\eta_0, \{x_1, \ldots, x_n\}, \epsilon, \lambda$
  1: Find $m \in (0, 1]$ such that $\forall$ neighboring $\eta_P, \eta_Q \in pset(\eta_0/m, n), D_\lambda(p(\theta|\eta_P)||p(\theta|\eta_Q)) \leq \epsilon$
  2: Sample $\theta \sim p(\theta|\eta')$ where $\eta' = \eta_0/m + \sum_{i=1}^n (S(x_i), 1)$

---

**Theorem 14.** *For any $\Delta$-bounded minimal exponential family with prior $\eta_0$ in the interior of E, any $\lambda > 1$, and any $\epsilon > 0$, there exists $m^* \in (0, 1]$ such that using $m \in (0, m^*]$ in Algorithm 3 will achieve $(\lambda, \epsilon)$-RDP.*

Theorems 13 and 14 can be interpreted as demonstrating that any RDP privacy level can be achieved by setting $r$ or $m$ arbitrarily close to zero. A small $r$ implies a weak contribution from the data, while a small $m$ implies a strong prior that outweighs the contribution from the data. Setting $r = 1$ and $m = 1$ reduces to Algorithm 1, in which a sample is released from the true posterior without any modifications for privacy.

We have not yet specified how to find the appropriate values of $r$ or $m$, and the condition requires checking the supremum of divergences across the possible *pset* range of parameters arising as posteriors. However, with an additional assumption this supremum of divergences can be efficiently computed.

**Theorem 15.** *Let $e(\eta_P, \eta_Q, \lambda) = D_\lambda(p(\theta|\eta_P)||p(\theta|\eta_Q))$. For a fixed $\lambda$ and fixed $\eta_P$, the function $e$ is a convex function over $\eta_Q$.*

*If for any direction $v \in Diff$, the function $g_v(\eta) = v^\intercal \nabla^2 C(\eta) v$ is convex over $\eta$, then for a fixed $\lambda$, the function $f_\lambda(\eta_P) = \sup_{\eta_Q r-neighboring \, \eta_P} e(\eta_P, \eta_Q, \lambda)$ is convex over $\eta_P$ in the directions spanned by $Diff$.*

**Corollary 2.** *The Beta-Bernoulli system satisfies the conditions of Theorem 15 since the functions $g_v(\eta)$ have the form $(v^{(1)})^2(\psi_1(\eta^{(1)}) + \psi_1(\eta^{(2)} - \eta^{(1)}))$, and $\psi_1$ is the digamma function. Both pset and $Diff$ are defined as convex sets. The expression $\sup_{r-neighboring \, \eta_P, \eta_Q \in pset(\eta_0, n)} D_\lambda(p(\theta|\eta_P)||p(\theta|\eta_Q))$ is therefore equivalent to the maximum of $D_\lambda(p(\theta|\eta_P)||p(\theta|\eta_Q))$ where $\eta_P \in \eta_0 + \{(0, n), (n, n)\}$ and $\eta_Q \in \eta_P \pm (r, 0)$.*

*The higher dimensional Dirichlet-Categorical system also satsifies the conditions of Theorem 15. This result is located in the appendix.*

We can do a binary search over $(0, 1]$ to find an appropriate value of $r$ or $m$. At each candidate value, we only need to consider the boundary situations to evaluate whether this value achieves the desired RDP privacy level. These boundary situations depend on the choice of model, and not the data size $n$. For example, in the Beta-Bernoulli system, evaluating the supremum involves calculating the Rényi diverengence across at most 4 pairs of distributions, as in Corollary 2. In the $d$ dimensional Dirichlet-Categorical setting, there are $O(d^3)$ distribution pairs to evaluate.

Eventually, the search process is guaranteed to find a non-zero choice for $r$ or $m$ that achieves the desired privacy level, although the utility optimality of this choice is not guaranteed. If stopped early and none of the tested candidate values satisfy the privacy constraint, the analyst can either continue to iterate or decide not to release anything.

**Extensions.** These methods have convenient privacy implications to the settings where some data is public, such as after a data breach, and for releasing a statistical query. They can also be applied to non-$\Delta$-bounded exponential families with some caveats. These additional results are located in the appendix.

# 4 RDP for Generalized Linear Models with Gaussian Prior

In this section, we reinterpret some existing algorithms in [13] in the light of RDP, and use ideas from [13] to provide new RDP algorithms for posterior sampling for a subset of generalized linear models with Gaussian priors.

## 4.1 Background: Generalized Linear Models (GLMs)

The goal of generalized linear models (GLMs) is to predict an outcome $y$ given an input vector $x$; $y$ is assumed to be generated from a distribution in the exponential family whose mean depends on $x$ through $\mathbb{E}[y|x] = g^{-1}(w^\top x)$, where $w$ represents the weight of linear combination of $x$, and $g$ is called the link function. For example, in logistic regression, the link function $g$ is logit and $g^{-1}$ is the sigmoid function; and in linear regression, the link functions is the identity function. Learning in GLMs means learning the actual linear combination $w$.

Specifically, the likelihood of $y$ given $x$ can be written as $p(y|w,x) = h(y)\exp\left(yw^\top x - A(w^\top x)\right)$, where $x \in \mathcal{X}$, $y \in \mathcal{Y}$, $A$ is the log-partition function, and $h(y)$ the scaling constant. Given a dataset $D = \{(x_1, y_1), \ldots, (x_n, y_n)\}$ of $n$ examples with $x_i \in \mathcal{X}$ and $y_i \in \mathcal{Y}$, our goal is to learn the parameter $w$. Let $p(D|w)$ denote $p(\{y_1, \ldots, y_n\}|w, \{x_1, \ldots, x_n\}) = \prod_{i=1}^n p(y_i|w, x_i)$. We set the prior $p(w)$ as a multivariate Gaussian distribution with covariance $\Sigma = (n\beta)^{-1}I$, i.e., $p(w) \sim \mathcal{N}(0, (n\beta)^{-1}I)$. The posterior distribution of $w$ given $D$ can be written as

$$p(w|D) = \frac{p(D|w)p(w)}{\int_{\mathbb{R}^d} p(D|w')p(w')dw'} \propto \exp\left(-\frac{n\beta\|w\|^2}{2}\right)\prod_{i=1}^n p(y_i|w, x_i). \tag{6}$$

## 4.2 Mechanisms and Privacy Guarantees

First, we introduce some assumptions that characterize the subset of GLMs and the corresponding training data on which RDP can be guaranteed.

**Assumption 1.** *(1). $\mathcal{X}$ is a bounded domain such that $\|x\|_2 \leq c$ for all $x \in \mathcal{X}$, and $x_i \in \mathcal{X}$ for all $(x_i, y_i) \in D$. (2). $\mathcal{Y}$ is a bounded domain such that $\mathcal{Y} \subseteq [y_{\min}, y_{\max}]$, and $y_i \in \mathcal{Y}$ for all $(x_i, y_i) \in D$.. (3). $g^{-1}$ has bounded range such that $g^{-1} \in [\gamma_{\min}, \gamma_{\max}]$. Then, let $B = \max\{|y_{\min} - \gamma_{\max}|, |y_{\max} - \gamma_{\min}|\}$.*

**Example: Binary Regression with Bounded $\mathcal{X}$** Binary regression is used in the case where $y$ takes value $\mathcal{Y} = \{0, 1\}$. There are three common types of binary regression, logistic regression with $g^{-1}(w^\top x) = 1/(1 + \exp\left(-w^\top x\right))$, probit regression with $g^{-1}(w^\top x) = \Phi(w^\top x)$ where $\Phi$ is the Gaussian cdf, and complementary log-log regression with $g^{-1}(w^\top x) = 1 - \exp\left(-\exp\left(w^\top x\right)\right)$. In these three cases, $\mathcal{Y} = \{0, 1\}$, $g^{-1}$ has range $(0, 1)$ and thus $B = 1$. Moreover, it is often assumed for binary regression that any example lies in a bounded domain, i.e., $\|x\|_2 \leq c$ for $x \in \mathcal{X}$.

Now we establish the privacy guarantee for sampling directly from the posterior in (6) in Theorem 17. We also show that this privacy bound is tight for logistic regression; a detailed analysis is in Appendix.

**Theorem 16.** *Suppose we are given a GLM and a dataset $D$ of size $n$ that satisfies Assumption 1, and a Gaussian prior with covariance $\Sigma = (n\beta)^{-1}I$, then sampling with posterior in* (6) *satisfies $(\lambda, \frac{2c^2B^2}{n\beta}\lambda)$-RDP for all $\lambda \geq 1$.*

Notice that direct posterior sampling cannot achieve $(\lambda, \epsilon)$-RDP for arbitrary $\lambda$ and $\epsilon$. We next present Algorithm 4 and 5, as analogous to Algorithm 3 and 2 for exponential family respectively, that guarantee any given RDP requirement. Algorithm 4 achieves a given RDP level by setting a stronger prior, while Algorithm 5 by raising the temperature of the likelihood.

| **Algorithm 4** Concentrated Posterior | **Algorithm 5** Diffuse Posterior |
|---|---|
| **Require:** Dataset $D$ of size $n$; Gaussian prior with covariance $(n\beta_0)^{-1}I$; $(\lambda, \epsilon)$. | **Require:** Dataset $D$ of size $n$; Gaussian prior with covariance $(n\beta)^{-1}I$; $(\lambda, \epsilon)$. |
| 1: Set $\beta = \max\{\frac{2c^2B^2\lambda}{n\epsilon}, \beta_0\}$ in (6).<br>2: Sample $w \sim p(w|D)$ in (6). | 1: Replace $p(y_i|w, x_i)$ with $p(y_i|w, x_i)^\rho$ in (6) where $\rho = \min\{1, \sqrt{\frac{\epsilon n\beta}{2c^2B^2\lambda}}\}$.<br>2: Sample $w \sim p(w|D)$ in (6). |

It follows directly from Theorem 17 that under Assumption 1, Algorithm 4 satisfies $(\lambda, \epsilon)$-RDP.

**Theorem 17.** *Suppose we are given a GLM and a dataset $D$ of size $n$ that satisfies Assumption 1, and a Gaussian prior with covariance $\Sigma = (n\beta)^{-1}I$, then Algorithm 5 guarantees $(\lambda, \epsilon)$-RDP. In fact, it guarantees $(\tilde{\lambda}, \frac{\epsilon}{\lambda}\tilde{\lambda})$-RDP for any $\tilde{\lambda} \geq 1$.*

## 5 Experiments

In this section, we present the experimental results for our proposed algorithms for both exponential family and GLMs. Our experimental design focuses on two goals – first, analyzing the relationship between $\lambda$ and $\epsilon$ in our privacy guarantees and second, exploring the privacy-utility trade-off of our proposed methods in relation to existing methods.

### 5.1 Synthetic Data: Beta-Bernoulli Sampling Experiments

In this section, we consider posterior sampling in the Beta-Bernoulli system. We compare three algorithms. As a baseline, we select a modified version of the algorithm in [10], which privatizes the sufficient statistic of the data to create a privatized posterior. Instead of Laplace noise that is used by[10], we use Gaussian noise to do the privatization; [14] shows that if Gaussian noise with variance $\sigma^2$ is added, then this offers an RDP guarantee of $(\lambda, \lambda\frac{\Delta^2}{\sigma^2})$ for $\Delta$-bounded exponential families. We also consider the two algorithms presented in Section 3.1 – Algorithm 2 and 3; observe that Algorithm 1 is a special case of both. 500 iterations of binary search were used to select $r$ and $m$ when needed.

**Achievable Privacy Levels.** We plot the $(\lambda, \epsilon)$-RDP parameters achieved by Algorithms 2 and 3 for a few values of $r$ and $m$. These parameters are plotted for a prior $\eta_0 = (6, 18)$ and the data size $n = 100$ which are selected arbitrarily for illustrative purposes. We plot over six values $\{0.1, 0.3, 0.5, 0.7, 0.9, 1\}$ of the scaling constants $r$ and $m$. The results are presented in Figure 1. Our primary observation is the presence of the vertical asymptotes for our proposed methods. Recall that any privacy level is achievable with our algorithms given small enough $r$ or $m$; these plots demonstrate the interaction of $\lambda$ and $\epsilon$. As $r$ and $m$ decrease, the $\epsilon$ guarantees improve at each $\lambda$ and even become finite at larger orders $\lambda$, but a vertical asymptote still exists. The results for the baseline are not plotted: it achieves RDP along any line of positive slope passing through the origin.

**Privacy-Utility Tradeoff.** We next evaluate the privacy-utility tradeoff of the algorithms by plotting $KL(P||\mathcal{A})$ as a function of $\epsilon$ with $\lambda$ fixed, where $P$ is the true posterior and $\mathcal{A}$ is the output distribution of a mechanism. For Algorithms 2 and 3, the KL divergence can be evaluated in closed form. For the Gaussian mechanism, numerical integration was used to evaluate the KL divergence integral. We have arbitrarily chosen $\eta_0 = (6, 18)$ and data set $\mathbf{X}$ with 100 total trials and 38 successful trials. We have plotted the resulting divergences over a range of $\epsilon$ for $\lambda = 2$ in (a) and for $\lambda = 15$ in (b) of Figure 2. When $\lambda = 2 < \lambda^*$, both Algorithms 2 and 3 reach zero KL divergence once direct sampling is possible. The Gaussian mechanism must always add nonzero noise. As $\epsilon \to 0$, Algorithm 3 approaches a point mass distribution heavily penalized by the KL divergence. Due to its projection step, the Gaussian Mechanism follows a bimodal distribution as $\epsilon \to 0$. Algorithm 2 degrades to the prior, with modest KL divergence. When $\lambda = 15 > \lambda^*$, the divergences for Algorithms 2 and 3 are bounded away from 0, while the Gaussian mechanism still approaches the truth as $\epsilon \to \infty$. In a non-private setting, the KL divergence would be zero.

Finally, we plot $\log p(\mathbf{X}_H|\theta)$ as a function of $\epsilon$, where $\theta$ comes from one of the mechanisms applied to $\mathbf{X}$. Both $\mathbf{X}$ and $\mathbf{X}_H$ consist of 100 Bernoulli trials with proportion parameter $\rho = 0.5$. This

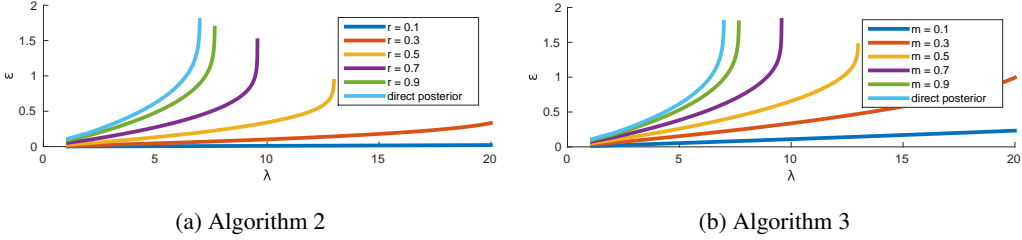

(a) Algorithm 2　　　　　　　　　　　　　　　(b) Algorithm 3

Figure 1: Illustration of Potential $(\lambda, \epsilon)$-RDP Curves for Exponential Family Sampling.

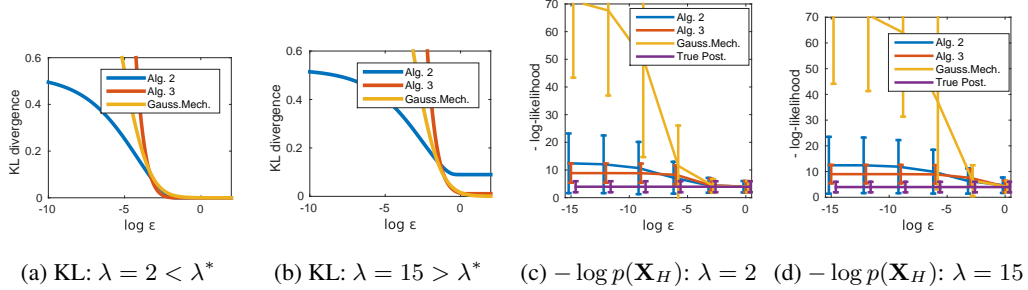

(a) KL: $\lambda = 2 < \lambda^*$　(b) KL: $\lambda = 15 > \lambda^*$　(c) $-\log p(\mathbf{X}_H)$: $\lambda = 2$　(d) $-\log p(\mathbf{X}_H)$: $\lambda = 15$

Figure 2: Exponential Family Synthetic Data Experiments.

experiment was run 10000 times, and we report the mean and standard deviation. Similar to the previous section, we have a fixed prior of $\eta_0 = (6, 18)$. The results are shown for $\lambda = 2$ in (c) and for $\lambda = 15$ in (d) of 2. These results agree with the limit behaviors in the KL test. This experiment is more favorable for Algorithm 3, as it degrades only to the log likelihood under the mode of the prior. In this plot, we have included sampling from the true posterior as a non-private baseline.

## 5.2　Real Data: Bayesian Logistic Regression Experiments

We now experiment with Bayesian logistic regression with Gaussian prior on three real datasets. We consider three algorithms – Algorithm 4 and 5, as well as the OPS algorithm proposed in [18] as a sanity check. OPS achieves pure differential privacy when the posterior has bounded support; for this algorithm, we thus truncate the Gaussian prior to make its support the $L_2$ ball of radius $c/\beta$, which is the smallest data-independent ball guaranteed to contain the MAP classifier.

**Achievable Privacy Levels.** We consider the achievable RDP guarantees for our algorithms and OPS under the same set of parameters $\beta$, $c$, $\rho$ and $B = 1$. [18] shows that with the truncated prior, OPS guarantees $\frac{4c^2\rho}{\beta}$-differential privacy, which implies $(\lambda, \frac{4c^2\rho}{\beta})$-RDP for all $\lambda \in [1, \infty]$; whereas our algorithm guarantees $(\lambda, \frac{2c^2\rho^2}{n\beta}\lambda)$-RDP for all $\lambda \geq 1$. Therefore our algorithm achieves better RDP guarantees at $\lambda \leq \frac{2n}{\rho}$, which is quite high in practice as $n$ is the dataset size.

**Privacy-Utility: Test Log-Likelihood and Error.** We conduct Bayesian logistic regression on three real datasets: Abalone, Adult and MNIST. We perform binary classification tasks: abalones with less than 10 rings vs. the rest for Abalone, digit 3 vs. digit 8 for MNIST, and income $\leq 50K$ vs. $> 50K$ for Adult. We encode all categorical features with one-hot encoding, resulting in 9 dimensions for Abalone, 100 dimensions for Adult and 784 dimensions in MNIST. We then scale each feature to range from $[-0.5, 0.5]$, and normalize each example to norm 1. $1/3$ of the each dataset is used for testing, and the rest for training. Abalone has 2784 training and 1393 test samples, Adult has 32561 and 16281, and MNIST has 7988 and 3994 respectively.

For all algorithms, we use an original Gaussian prior with $\beta = 10^{-3}$. The posterior sampling is done using slice sampling with 1000 burn-in samples. Notice that slice sampling does not give samples from the exact posterior. However, a number of MCMC methods are known to converge in total variational distance in time polynomial in the data dimension for log-concave posteriors (which is the case here) [17]. Thus, provided that the burn-in period is long enough, we expect the induced

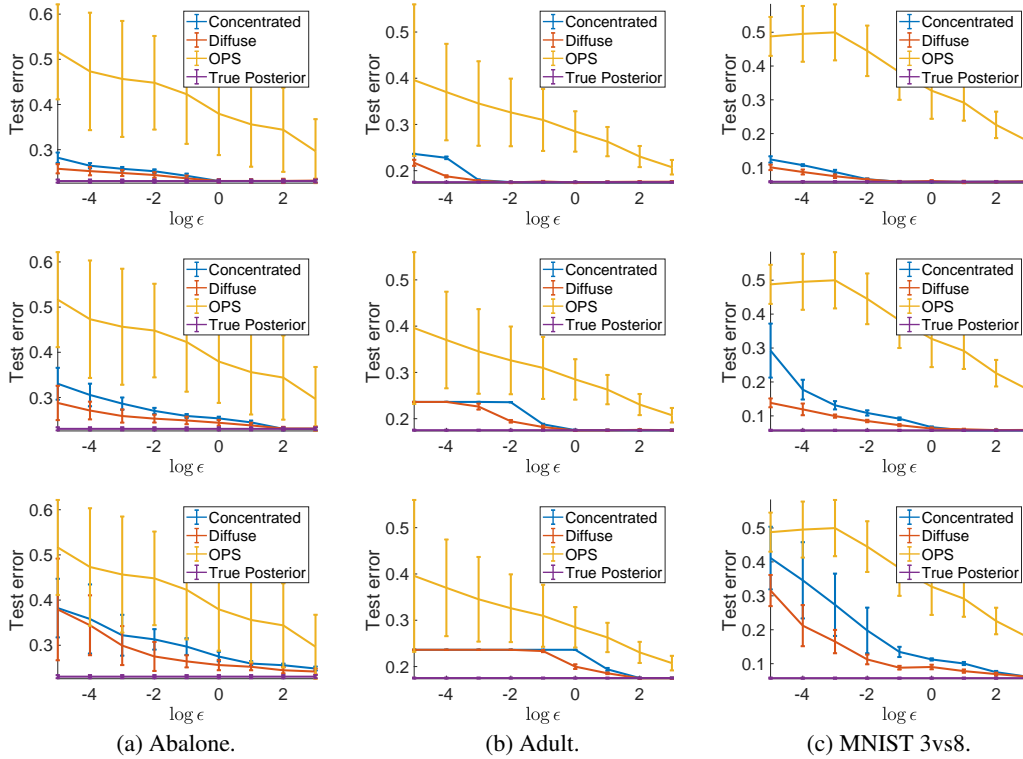

Figure 3: Test error vs. privacy parameter $\epsilon$. $\lambda = 1, 10, 100$ from top to bottom.

distribution to be quite close, and we leave an exact RDP analysis of the MCMC sampling as future work. For privacy parameters, we set $\lambda = 1, 10, 100$ and $\epsilon \in \{e^{-5}, e^{-4}, \ldots, e^3\}$. Figure 3 shows the test error averaged over 50 repeated runs. More experiments for test log-likelihood presented in the Appendix.

We see that both Algorithm 4 and 5 achieve lower test error than OPS at all privacy levels and across all datasets. This is to be expected, since OPS guarantees pure differential privacy which is stronger than RDP. Comparing Algorithm 4 and 5, we can see that the latter always achieves better utility.

## 6 Conclusion

The inherent randomness of posterior sampling and the mitigating influence of a prior can be made to offer a wide range of privacy guarantees. Our proposed methods outperform existing methods in specific situations. The privacy analyses of the mechanisms fit nicely into the recently introduced RDP framework, which continues to present itself as a relaxation of DP worthy of further investigation.

### Acknowledgements

This work was partially supported by NSF under IIS 1253942, ONR under N00014-16-1-2616, and a Google Faculty Research Award.

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
