[Supplementary Material · nips_2017_supplement.pdf]

# A  Appendix

## A.1   Background: Exponential Families

This section will give a in-depth explanation of exponential families and the properties of them we exploit in our analysis.

An exponential family is a family of probability distributions over $x \in \mathcal{X}$ indexed by the parameter $\theta \in \Theta \subseteq \mathbb{R}^d$ that can be written in this canonical form for some choice of functions $h : \mathcal{X} \to \mathbb{R}$, $S : \mathcal{X} \to \mathbb{R}^d$, and $A : \Theta \to \mathbb{R}$:

$$p(x_1, \ldots, \mathbf{x}_n | \theta) = (\prod_{i=1}^{n} h(x_i)) \exp\left( (\sum_{i=1}^{n} S(x_i)) \cdot \theta - n \cdot A(\theta) \right) . \tag{1}$$

We call $h$ the base measure, $S$ the sufficient statistics of $x$, and $A$ as the log-partition function of this family. Note that the data $\{x_1, \ldots, \mathbf{x}_n\}$ interact with the parameter $\theta$ solely through the dot product of $\theta$ and the sum of their sufficient statistics. When the parameter $\theta$ is used in this dot product unmodified (as in (1)), we call this a natural parameterization. Our analysis will be restricted to the families that satisfy the following two properties:

**Definition A.1.** *An exponential family is minimal if the coordinates of the function $S$ are not almost surely linearly dependent, and the interior of $\Theta$ is non-empty.*

**Definition A.2.** *For any for $\Delta \in \mathbb{R}$, an exponential family is $\Delta$-bounded if*

$$\Delta \geq \sup_{x,y \in \mathcal{X}} ||S(x) - S(y)||. \tag{2}$$

*This constraint can be relaxed with some caveats explored in the appendix.*

When a family is minimal, the log-partition function $A$ has many interesting characteristics. It can be defined as $A(\theta) = \log \int_{\mathcal{X}} h(x) \exp\left( S(x) \cdot \theta \right) dx$, and serves to normalize the distribution. Its derivatives form the cumulants of the distribution, that is to say $\nabla A(\theta) = \kappa_1 = \mathbb{E}_{x|\theta}[S(x)]$ and $\nabla^2 A(\theta) = \kappa_2 = \mathbb{E}_{x|\theta}[(S(x) - \kappa_1)(S(x) - \kappa_1)^{\mathsf{T}}]$. This second cumulant is also the covariance of $S(x)$, which demonstrates that $A(\theta)$ must be a convex function since covariances must be positive semidefinite.

In Bayesian data analysis, we are interested in finding our posterior distribution over the parameter $\theta$ that generated the data. We must introduce a prior distribution $p(\theta|\eta)$ to describe our initial beliefs on $\theta$, where $\eta$ is a parameterization of our family of priors.

$$p(\theta|x_1, \ldots, x_n, \eta) \propto p(x_1, \ldots, x_n|\theta)p(\theta|\eta) \tag{3}$$

$$\propto (\prod_{i=1}^{n} h(x_i)) \exp\left( (\sum_{i=1}^{n} S(x_i)) \cdot \theta - n \cdot A(\theta) \right) p(\theta|\eta) \tag{4}$$

$$\propto \exp\left( (\sum_{i=1}^{n} S(x_i), n) \cdot (\theta, -A(\theta)) \right) p(\theta|\eta) \tag{5}$$

$$\tag{6}$$

Notice that we can ignore the $(\prod_{i=1}^{n} h(x_i))$ as it is a constant that will be normalized out. If we let our prior take the form of another exponential family $p(\theta|\eta) = \exp\left( T(\theta) \cdot \eta - B(\eta) \right)$ where $T(\theta) = (\theta, -A(\theta))$ and $B(\eta) = \log \int_{\Theta} \exp\left( T(\theta) \cdot \eta \right) d\theta$, the we can perform these manipulations,

$$p(\theta|x_1, \ldots, x_n, \eta) \propto \exp\left( (\sum_{i=1}^{n} S(x_i), n) \cdot T(\theta) + \eta \cdot T(\theta) - B(\eta) \right) \tag{7}$$

$$\propto \exp\left( \left( \eta + (\sum_{i=1}^{n} S(x_i), n) \right) \cdot T(\theta) - B(\eta) \right) \tag{8}$$

and see that expression (8) can be written as

$$p(\theta|\eta') = \exp\left(T(\theta) \cdot \eta' - C(\eta')\right) \tag{9}$$

where $\eta' = \eta + \sum_{i=1}^{n}(S(x_i), 1)$ and $C(\eta')$ is chosen such that the distribution is normalized.

This family of posteriors is precisely the same exponential family that we chose for our prior. We call this a conjugate prior, and it offers us an efficient way of finding the parameter of our posterior: $\eta_{posterior} = \eta_{prior} + \sum_{i=1}^{n}(S(x_i), 1)$. Within this family, $T(\theta)$ forms the sufficient statistics of $\theta$, and the derivatives of $C(\eta)$ give the cumulants of these sufficient statistics.

**Beta-Bernoulli System.** A specific example of an exponential family that we will be interested in is the Beta-Bernoulli system, where an individual's data is a single i.i.d. bit modeled as a Bernoulli variable with parameter $\rho$, along with a Beta conjugate prior.

$$p(x_1, \ldots, \mathbf{x}_n|\rho) = \prod_{i=1}^{n} \rho^{x_i}(1-\rho)^{1-x_i} \tag{10}$$

Letting $\theta = \log(\frac{\rho}{1-\rho})$ and $A(\theta) = \log(1 + \exp(\theta)) = -\log(1-\rho)$, we can rewrite the equation as follows:

$$p(x_1, \ldots, \mathbf{x}_n|\rho) = \prod_{i=1}^{n}(\frac{\rho}{1-\rho})^{x_i}(1-\rho) \tag{11}$$

$$= \exp\left(\sum_{i=1}^{n} x_i \log(\frac{\rho}{1-\rho}) + \log(1-\rho)\right) \tag{12}$$

$$= \exp\left((\sum_{i=1}^{n} x_i) \cdot \theta - A(\theta)\right). \tag{13}$$

This system satisfies the properties we require, as this natural parameterization with $\theta$ is both minimal and $\Delta$-bounded for $\Delta = 1$.

As our mechanisms are interested mainly in the posterior, the rest of this section will be written with respect the family specified by equation (9).

Now that we have the notation for our distributions, we can write out the expression for the Rényi divergence of two posterior distributions $P$ and $Q$ (parameterized by $\eta_P$ and $\eta_Q$) from the same exponential family. This expression allows us to directly compute the Rényi divergences of posterior sampling methods, and forms the crux of the analysis of our exponential family mechanisms.

**Observation A.3.** *Let $P$ and $Q$ be two posterior distributions from the same exponential family that are parameterized by $\eta_P$ and $\eta_Q$. Then,*

$$D_\lambda(P||Q) = \frac{1}{\lambda - 1}\log\left(\int_{\Theta} P(\theta)^\lambda Q(\theta)^{1-\lambda}d\theta\right) = \frac{C(\lambda\eta_P + (1-\lambda)\eta_Q) - \lambda C(\eta_P)}{\lambda - 1} + C(\eta_Q). \tag{14}$$

To help analyze the implication of equation (14) for Rényi Differential Privacy, we define some sets of prior/posterior parameters $\eta$ that arise in our analysis.

**Definition A.4.** *We say a posterior parameter $\eta$ is normalizable if $C(\eta) = \log\int_{\Theta} exp\left(T(\theta) \cdot \eta\right) d\theta$ is finite.*

*Let $E$ denote the set of all normalizable $\eta$ for the conjugate prior family.*

**Definition A.5.** *Let $pset(\eta_0, n)$ be the convex hull of all parameters $\eta$ of the form $\eta_0 + n(S(x), 1)$ for $x \in \mathcal{X}$. When $n$ is an integer this represents the hull of possible posterior parameters after observing $n$ data points starting with the prior $\eta_0$.*

**Definition A.6.** *Let $Diff$ be the difference set for the family, where $Diff$ is the convex hull of all vectors of the form $(S(x) - S(y), 0)$ for $x, y \in \mathcal{X}$.*

**Definition A.7.** *Two posterior parameters $\eta_1$ and $\eta_2$ are neighboring iff $\eta_1 - \eta_2 \in Diff$. They are r-neighboring iff $(\eta_1 - \eta_2)/r \in Diff$.*

## A.2 Extension: Public Data for Exponential Families

The use of a conjugate prior makes the interaction of observed data versus the prior easy to see. The prior $\eta_0$ can be expressed as $(\alpha\chi, \alpha)$, where $\chi$ is a vector expressing the average sufficient statistics of pseudo-observations and $\alpha$ represents a count of these pseudo-observations. After witnessing the $n$ data points, the posterior becomes a prior that has averaged the data sufficient statistics into a new $\chi$ and added $n$ to $\alpha$.

If the data analyst had some data in addition to $\mathbf{X}$ that was not privacy sensitive, perhaps from a stale data set for which privacy requirements have lapsed, then this data can be used to form a better prior for the analysis.

Not only would this improve utility by adding information that can be fully exploited, it would also in most cases improve the privacy guarantees as well. A stronger prior, especially a prior farther from the boundaries where $C(\eta)$ becomes infinite, will lead to smaller Rényi divergences. This is effectively the same behavior as the Concentrated Sampling mechanism, which scales the prior to imagine more pseudo-observations had been seen. This also could apply to settings in which the analyst can adaptively pay to receive non-private data, since this method will inform us once our prior formed from this data becomes strong enough to sample directly at our desired RDP level.

This also carries another privacy implication for partial data breaches. If an adversary learns the data of some individuals in the data set, the Direct Sampling mechanism's privacy guarantee for the remaining individuals can actually improve. Any contributions of the affected individuals to the posterior become in effect yet more public data placed in the prior. The privacy analysis and subsequent guarantees will match the setting in which this strengthened prior was used.

## A.3 Extension: Releasing the result of a Statistical Query

Here we are given a sensitive database $\mathbf{X} = \{x_1, \ldots, x_n\}$ and a predicate $\phi(\cdot)$ which maps each $x_i$ into the interval $[0, 1]$. Our goal is to release a Rényi DP approximation to the quantity: $F(\mathbf{X}) = \frac{1}{n}\sum_{i=1}^{n}\phi(x_i)$.

Observe that directly releasing $F(\mathbf{X})$ is neither DP nor Rényi DP, since this is a deterministic algorithm; our goal is to release a random sample from a suitable distribution so that the output is as close to $F(\mathbf{X})$ as possible.

The task of releasing a privatized result of a statistical query can be embedded into our Beta-Bernoulli system. This allows the privatized statistical query release to be done using either Algorithm 2 or Algorithm 3.

We can extend the Beta-Bernoulli model to allow the sufficient statistics $S(x)$ to range over the interval $[0, 1]$ instead of just the discrete set $\{0, 1\}$. This alteration still results in a $\Delta$-bounded exponential family, and the privacy results hold.

The sampled posterior will be a Beta distribution that will concentrate around the mean of the data observations and the pseudo-observations of the prior. The process is described in the Beta-Sampled Statistical Query algorithm. The final transformation maps the natural parameter $\theta \in (-\infty, \infty)$ onto the mean of the distribution $\rho \in (0, 1)$.

## A.4 Proofs of Exponential Family Sampling Theorems

Our proofs will make extensive use of the definitions laid out in Section A.1. We will however need an additional definition for a modified version of $pset$, and as well the set of possible updates to the posterior parameter that might arise from the data.

---
**Algorithm 1** Beta-Sampled Statistical Query
---
**Require:** $\eta_0, \{x_1, \ldots, x_n\}, f, \epsilon, \lambda$
 1: Compute $\mathbf{X}_f = \{f(x_1), \ldots, f(x_n)\}$.
 2: Sample $\theta$ via Algorithm 2 or Algorithm 3 applied to $\mathbf{X}_f$ with $\eta_0, \epsilon$, and $\lambda$.
 3: Release $\rho = \frac{\exp(\theta)}{1+\exp(\theta)}$.
---

**Definition A.8.** *Let* $lpset(\eta_0, n, b) = pset(\eta_0, n) + bDiff$. *This is the set of posterior parameters that are $b$-neighboring at least one of the elements of* $pset(\eta_0, n)$

**Definition A.9.** *Let $U$ be the set of posterior updates for an exponential family, where $U$ is the convex hull of all vectors of the form* $(S(x), 1)$ *for* $x, y \in \mathcal{X}$.

We begin by noting that observing a data set when starting at a normalizable prior $\eta_0$ must result in a normalizable posterior parameter $\eta'$.

**Observation 1.** *In a minimal exponential family, for any prior parameter $\eta_0$, any $n > 0$, and any posterior update, every possible posterior parameter in the set $\eta_0 + nU$ is also normalizable. As $C(\eta)$ must be a convex function for minimal families, this must apply to positive non-integer values of $n$ as well.*

With this observation, we are ready to prove our result on the conditions under which sampling from our posterior gives a finite $(\lambda, \epsilon)$-RDP guarantee.

**Theorem A.10.** *For a $\Delta$-bounded minimal exponential family of distributions $p(x|\theta)$ with continuous log-partition function $A(\theta)$, there exists $\lambda^* \in (1, \infty]$ such Algorithm 1 achieves $(\lambda, \epsilon(\eta_0, n, \lambda))$-RDP for $\lambda < \lambda^*$.*

*$\lambda^*$ is the supremum over all $\lambda$ such that all $\eta$ in the set $\eta_0 + (\lambda - 1)Diff$ are normalizable.*

PROOF:

Algorithm 1 samples directly from the posterior $\eta_{post} = \eta_0 + \sum_i (S(x_i), 1)$. When applied to neighboring data sets $\mathbf{X}$ and $\mathbf{X}'$, it selects posterior parameters that are neighboring.

The theorem can be reinterpreted as saying there exists $\lambda^*$ such that for $\lambda < \lambda^*$ we have

$$\sup_{\text{neighboring } \eta_P, \eta_Q \in pset(\eta_0, n)} D_\lambda(p(\theta|\eta_P)||p(\theta|\eta_Q)) < \infty. \tag{15}$$

For these two posteriors from the same exponential family, we can write out the Rényi divergence in terms of the log-partition function $C(\eta)$.

$$D_\lambda(p(\theta|\eta_P)||p(\theta|\eta_Q)) = \frac{C(\lambda\eta_P + (1-\lambda)\eta_Q) - \lambda C(\eta_P)}{\lambda - 1} + C(\eta_Q) \tag{16}$$

We wish to show that this is bounded above over all neighboring $\eta_P$ and $\eta_Q$ our mechanism might generate, and will do so by showing that $|C(\eta)|$ must be bounded every where it is applied in equation (16) if $\lambda < \lambda^*$. To find this bound, we will ultimately show each potential application of $C(\eta)$ lies within a closed subset of $E$, from which the continuity of $C$ will imply an upperbound.

Let's begin by observing that $\eta_P$ and $\eta_Q$ must lie within $pset(\eta_0, n)$ as they arise as posteriors for neighboring data sets $\mathbf{X}$ and $\mathbf{X}'$. The point $\eta_L = \lambda\eta_P + (1-\lambda)\eta_Q = \eta_P + (\lambda - 1)(\eta_P - \eta_Q)$ might not lie within $pset(\eta_0, n)$. However, we know $\eta_P - \eta_Q$ lies within $Diff$ and that $\eta_L - \eta_P$ is within $(\lambda - 1)Diff$. This means for any neighboring data sets, $\eta_P, \eta_Q$, and $\eta_L$ lie inside $lpset(\eta_0, n, \lambda - 1)$.

If $\lambda < \lambda^*$, then $\eta_0 + (\lambda - 1)Diff \subseteq E$. The set $\eta_0 + (\lambda - 1)Diff$ is potentially an open set, but the closure of this set must be within $E$ as well, since we can always construct $\lambda' \in (\lambda, \lambda^*)$ where $\eta_0 + (\lambda' - 1)Diff \subseteq E$, and the points inside $\eta_0 + (\lambda - 1)Diff$ can't converge to any point outside of $\eta_0 + (\lambda' - 1)Diff$.

Any point in $\eta \in lpset(\eta_0, n, \lambda - 1)$ can be broken down into three components using the definition of $lpset$: $\eta = \eta_0 + u + d$, where $u \in nU$ and $d \in (\lambda - 1)Diff$. For any point in this $lpset$, we

can therefore subtract off the component $u$ to reach a point in the set $\eta_0 + (\lambda - 1)Diff$. With Observation 1, we can conclude that $\eta$ is normalizable if $\eta - u$ is normalizable, and therefore the closure of $lpset(\eta_0, n, \lambda - 1)$ is a subset of $E$ if $\eta_0 + (\lambda - 1)Diff$ is a subset of $E$, which we have shown for $\lambda < \lambda^*$.

As $C(\eta)$ is a continuous function, we know that the supremum of $|C(\eta)|$ over the closure of $lpset(\eta_0, n, \lambda - 1)$ must be finite. Remember that for any neighboring data sets, $\eta_P, \eta_Q$, and $\eta_L$ are inside $lpset(\eta_0, n, \lambda - 1)$. Since $|C(\eta)|$ is bounded over this $lpset$, so too must our expression for $D_\lambda(p(\theta|\eta_P)||p(\theta|\eta_Q))$ in equation (16). Therefore there exists an upper-bound for the order $\lambda$ Rényi divergence across all pairs of posterior parameters selected by Algorithm 1 on neighboring data sets. This finite upper-bound provides a finite value for $\epsilon(\eta_0, n, \lambda)$ for which Algorithm 1 offers $(\lambda, \epsilon(\eta_0, n, \lambda))$-RDP .

$\square$

To prove our results for Algorithm 2 and Algorithm 3, we'll need an additional result that bounds the Rényi divergence in terms of the Hessian of the log-partition function and the distance between the two distribution parameters.

**Lemma 1.** *For $\lambda > 1$, if $||\nabla^2 C(\eta)|| < H$ over the set $\{\eta_P + x(\eta_P - \eta_Q)|x \in [-\lambda + 1, \lambda - 1]\}$, then*

$$D_\lambda(p(\theta|\eta_P)||p(\theta|\eta_Q)) \leq ||\eta_P - \eta_Q||^2 H \lambda \tag{17}$$

PROOF:

Define the function $g(x) = C(\eta_P + xv)$ where $x \in \mathbb{R}$ and $v = \eta_P - \eta_Q$. This allows us to rewrite the Rényi divergence as

$$D_\lambda(P||Q) = \frac{g(1 - \lambda) - \lambda g(0)}{\lambda - 1} + g(1) \tag{18}$$

Now we will replace $g$ with its first order Taylor expansion

$$g(x) = g(0) + xg'(0) + e(x) \tag{19}$$

where $e(x)$ is the approximation error term, satisfying $|e(x)| \leq x^2 \max_{y \in [-x,x]} g''(y)/2$.

This results in

$$D_\lambda(p(\theta|\eta_P)||p(\theta|\eta_Q)) = \frac{g(0) + (1 - \lambda)g'(0) + e(1 - \lambda) - \lambda g(0)}{\lambda - 1} + g(0) + g'(0) + e(1) \tag{20}$$

$$= -\frac{e(1 - \lambda)}{\lambda - 1} + e(1) \tag{21}$$

$$\leq \frac{(\lambda - 1)^2}{\lambda - 1} \max_{y \in [-\lambda+1, \lambda-1]} g''(y)/2 + \max_{y \in [-1,1]} g''(y)/2 . \tag{22}$$

Further, we can express $g''$ in terms of $C$ and $v$.

$$g''(y) = v^\mathsf{T} \nabla^2 C(\eta_P + yv)v \tag{23}$$

$$\leq ||\eta_P - \eta_Q||^2 ||\nabla^2 C(\eta_P + yv)|| \tag{24}$$

$$\leq ||\eta_P - \eta_Q||^2 H \tag{25}$$

Plugging in this bound on $g''$ gives the desired result.

$$D_\lambda(p(\theta|\eta_P)||p(\theta|\eta_Q)) \leq \frac{(\lambda-1)^2}{\lambda-1} \max_{y\in[-\lambda+1,\lambda-1]} g''(y)/2 + \max_{y\in[-1,1]} g''(y)/2 \tag{26}$$

$$\leq (\lambda-1)||\eta_P - \eta_Q||^2 H/2 + ||\eta_P - \eta_Q||^2 H/2 \tag{27}$$

$$\leq ||\eta_P - \eta_Q||^2 H\lambda/2 \tag{28}$$

$$\leq ||\eta_P - \eta_Q||^2 H\lambda \tag{29}$$

□

We will also make use of the following standard results about the Hessian of the log-partition function of minimal exponential families, given in (**?**) as Theorem 1.17 and Corollary 1.19 and rephrased for our purposes.

**Theorem 2.** *(Theorem 1.17 from (**?**)) The log-partition function $C(\eta)$ of a minimal exponential family is infinitely often differentiable at parameters $\eta$ in the interior of the normalizable set $E$.*

**Theorem 3.** *(Corollary 1.19 from (**?**)) For minimal exponential family, the Hessian of the log-partition function $\nabla^2 C(\eta)$ is nonsingular for every parameter $\eta$ in the interior of the normalizable set $E$.*

These results imply that the Hessian $\nabla^2 C(\eta)$ must exist and be continuous over $\eta$ in the interior of $E$, as well as having non-zero determinant.

**Theorem A.11.** *For any $\Delta$-bounded minimal exponential family with prior $\eta_0$ in the interior of $E$, any $\lambda > 1$, and any $\epsilon > 0$, there exists $r^* \in (0,1]$ such that using $r \in (0, r^*]$ in Algorithm 2 will achieve $(\lambda, \epsilon)$-RDP.*

PROOF:

Recall that Algorithm 2 uses the posterior parameter $\eta' = \eta_0 + r\sum_i^n(S(x), 1)$ where the data contribution has been scaled by $r$. Our first step of this proof is to show that there exists $r_0 \in (0,1]$ such that the order $\lambda$ Rényi divergences of the generated parameters are finite for $r < r_0$.

Similar to the proof of Theorem A.10, we will do so by creating a closed set where $C(\eta)$ is finite and that must contain $\eta_P, \eta_Q$, and $\eta_L$ for any choice of neighboring data sets. On neighboring data sets, this generates $r$-neighboring parameters $\eta_P$ and $\eta_Q$. The point $\eta_L = \lambda\eta_P + (1-\lambda)\eta_Q$ is therefore $r(\lambda-1)$-neighboring $\eta_P$. These points must be contained in the set $lpset(\eta_0, rn, r(\lambda-1)) = \eta_0 + rnU + r(\lambda-1)Diff$. For any point in this set, we can subtract off the component in $rnU$ to get to a modified prior that is $r(\lambda-1)$-neighboring $\eta_0$.

By the assumption that $\eta_0$ is in the interior of $E$, there exists $\delta > 0$ such that the ball $\mathcal{B}(\eta_0, \delta) \subseteq E$. For the choice $r_0 = \frac{\delta}{2(\lambda-1)\Delta}$, for any $r \in (0, r_0)$, the modified prior we constructed for each point in $lpset(\eta_0, rn, r(\lambda-1))$ is within distance $r(\lambda-1)\Delta$ of $\eta_0$ and therefore within $\mathcal{B}(\eta_0, \delta/2) \subset \mathcal{B}(\eta_0, \delta) \subseteq E$. Observation 1 then allows us to conclude that every point $\eta$ in $lpset(\eta_0, rn, r(\lambda-1))$ has an open neighborhood of radius $\delta_2$ where $C(\eta)$ is finite. This is enough to conclude that the closure of this $lpset$ must also lie entirely within $E$, and $C(\eta)$ is finite and continuous over this closed set. As in Theorem A.10, this suffices to show that the supremum of order $\lambda$ Rényi divergences on neighboring data sets is bounded above.

We have thus shown there exists $r_0$ where the $\epsilon$ of our $(\lambda, \epsilon)$-RDP guarantee is finite for $r < r_0$. However, our goal was to achieve a specific $\epsilon$ guarantee. Our proof of the existence of $r^*$ centers around the claim that there must exist a bound $H$ for the Hessian of $C(\eta)$ over all choices of $r \in [0, r_0)$.

We can construct the set $D = \cup_{r\in[0,r_0]} lpset(\eta_0, rn, r(\lambda-1))$, which will contain every possible $\eta_P, \eta_Q$, and $\eta_L$ that might arise from any pair neighboring data sets and any choice of $r$ in that interval. The previous argument still applies: each point in this union must have an open neighborhood of radius $\delta/2$ that is a subset of $E$. This is enough to conclude that closure of $D$ is also a subset of $E$. Theorem 2 implies $\nabla^2 C(\eta)$ exists and is continuous on the interior of $E$, and this further implies that there must exist $H$ such that for all $\eta$ in this closure we have $||\nabla^2 C(\eta)|| \leq H$.

For any value $r$, we know that $\eta_P$ and $\eta_Q$ are $r$-neighboring, so we know $||\eta_P - \eta_Q|| \leq r\Delta$. Since $D$ contains $lpset(\eta_0, rn, r(\lambda-1))$, the bound $H$ must apply for all $\eta$ in the set $\{\eta_P + x(\eta_P - \eta_Q)|x \in [-\lambda+1, \lambda-1]\}$. This allows us to use Lemma 1 to get the following expression:

$$D_\lambda(p(\theta|\eta_P)||p(\theta|\eta_Q)) \leq ||\eta_P - \eta_Q||^2 H\lambda \tag{30}$$

$$\leq r\Delta^2 H\lambda. \tag{31}$$

If we set $r^* = \frac{\epsilon}{\Delta^2 H\lambda}$, then for $r < r^*$ the order $\lambda$ Rényi divergence of Algorithm 2 is bounded above by $\epsilon$, which gives us the desired result.

□

The concentrated mechanism is a bit more subtle in how it reduces the influence of the data, and so we need this result modified from Lemmas 9 and 10 in the appendix of (**?**). These results are presented here in a way that matches our notation. It effectively states that if we start at a prior $\eta_0$ satisfy mild but technical regularity assumptions, then the Hessians $C(k\eta_0)$ must converge to zero as $k$ grows. In practical terms, this implies the covariance of our prior distribution must shrink as we increase the number of pseudo-observations.

**Definition A.12.** *Let $T_\eta^* = T(\mathrm{argmax}_{\theta \in \Theta} \eta \cdot T(\theta))$. This represents the mode of the sufficient statistics under the distribution $p(T(\theta)|\eta)$.*

**Lemma 4.** *(Lemma 9 from (**?**)) If $A(\theta)$ is continuously differentiable and $\eta_0$ is in the interior of $E$, then $\mathrm{argmax}_{\theta \in \Theta} \eta \cdot T(\theta)$ must be in the interior of $\Theta$.*

**Lemma 5.** *(Lemma 10 from (**?**)) If we have a minimal exponential family in which $A(\theta)$ is differentiable of all orders, there exists $\delta_1 > 0$ such that the ball $\mathcal{B}(\eta_0, \delta_1)$ is a subset of $E$, there exists $\delta_2 > 0$ and a bound $L$ such that all the seventh order partial derivatives of $A(\theta)$ on the set $D_{\eta_0, \delta_1, \delta_2} = \{\theta| \min_{\eta \in \mathcal{B}(\eta_0, \delta_1)} ||T(\theta) - T_\eta^*|| < \delta_2\}$ are bounded by $P$, and the determinant of $\nabla^2 A(\theta)$ is bounded away from zero on $D_{\eta_0, \delta_1, \delta_2}$, then there exists real number $V, K$ such that for $k > K$ we have*

$$\forall \eta \in \mathcal{B}(\eta_0, \delta_1) \ ||\nabla^2 C(k\eta)|| < \frac{V}{k}. \tag{32}$$

**Theorem A.13.** *For any $\Delta$-bounded minimal exponential family with prior $\eta_0$ in the interior of $E$, for any $\lambda > 1$, and any $\epsilon > 0$, there exists $m^* \in (0, 1]$ such that using $m \in (0, m^*]$ in Algorithm 3 will achieve $(\lambda, \epsilon)$-RDP.*

PROOF:

For a fixed value of $m$, recall that Algorithm 3 selects the posterior parameter $\eta' = m^{-1}\eta_0 + \sum_{i=1}^n (S(x_i), 1)$. For neighboring data sets $\mathbf{X}$ and $\mathbf{X}'$, the selected posterior parameters $\eta_P, \eta_Q$, and $\eta_L = \lambda\eta_P + (1-\lambda)\eta_Q$ lie within $lpset(m^{-1}\eta_0, n, \lambda - 1) = m^{-1}\eta_0 + nU + (\lambda - 1)Diff$.

We start by showing that the conditions of Lemma 5 are met. As we assumed $\eta_0$ is in the interior of $E$, there exists $\delta_1 > 0$ such that we have the ball $\mathcal{B}(\eta_0, \delta_1) \subseteq E$. By Theorem 2, the log-partition function of the data likelihood $A(\theta)$ is differentiable of all orders, and Theorem 3 tells us that the Hessian $\nabla^2 A(\theta)$ is non-singular with non-zero determinant on the interior of $\Theta$. This permits the application of Lemma 4, offering a mapping from $\eta$ in the interior of $E$ to their mode $T_\eta^*$ corresponding to a parameter $\theta$ in the interior of $\Theta$. Knowing that $A(\theta)$ is infinitely differentiable on the interior of $\Theta$ further implies that the seventh order derivatives are well-behaved in a neighborhood around each mode resulting from this mapping. This provides the rest of the requirements for Lemma 5.

Therefore there exists $V$ and $K$ such that the following holds

$$\forall \eta \in \mathcal{B}(\eta_0, \delta_1) : ||\nabla^2 C(k\eta)|| \leq \frac{V}{k}. \tag{33}$$

We wish to show that $||\nabla^2 C(\eta)||$ must be bounded on the expanded set $lpset(m^{-1}\eta_0, n, \lambda - 1) = m^{-1}\eta_0 + nU + (\lambda - 1)Diff$, and will do so by showing that for small enough $m$ we can use equation (33) to bound the Hessians.

Let $\alpha(\eta)$ denote the last coordinate of $\eta$. This represents the pseudo-observation count of this parameter, and notice that $\forall u \in U : \alpha(u) = 1$ and $\forall v \in Diff : \alpha(v) = 0$. We are going to analyze

the scaled set $c_m \cdot lpset(m^{-1}\eta_0, n, \lambda - 1)$ where $c_m$ is a positive scaling constant that will depend on $m$.

$$c_m \cdot lpset(m^{-1}\eta_0, n, \lambda - 1) = c_m m^{-1}\eta_0 + c_m n U + c_m(\lambda - 1)Diff \tag{34}$$

.

For each $\eta$ in this $c_m \cdot lpset$, we have

$$\alpha(\eta) = c_m m^{-1}\alpha(\eta_0) + c_m n \cdot 1 + c_m(\lambda - 1) \cdot 0 = c_m(m^{-1}\alpha(\eta_0) + n) . \tag{35}$$

Setting $c_m = \frac{\alpha(\eta_0)}{m^{-1}\alpha(\eta_0)+n}$ thus guarantees that for all $\eta$ in $c_m \cdot lpset(m^{-1}\eta_0, n, \lambda - 1)$ we have $\alpha(\eta) = \alpha(\eta_0)$. We want to know how far the points in this $c_m \cdot lpset$ are from $\eta_0$, so we simply subtract $\eta_0$ to get a set $D_m$ of vectors. These offset vectors have the form $c_m \cdot lpset(m^{-1}\eta_0, n, \lambda - 1) - \eta_0$ and therefore lie in the set

$$D_m = (c_m m^{-1} - 1)\eta_0 + c_m n U + c_m(\lambda - 1)Diff. \tag{36}$$

Using our expression of $c_m$ as a function of $m$, we can see the following limiting behavior:

$$\lim_{m \to 0} c_m = \lim_{m \to 0} \frac{\alpha(\eta_0)}{m^{-1}\alpha(\eta_0) + n} = 0 \tag{37}$$

$$\lim_{m \to 0} c_m m^{-1} - 1 = \lim_{m \to 0} \frac{m^{-1}\alpha(\eta_0)}{m^{-1}\alpha(\eta_0) + n} - 1 = 1 - 1 = 0. \tag{38}$$

These limits lets us take the limit of the size of the vectors in $D_m$ as $m \to 0$:

$$\lim_{m \to 0} \sup_{v \in D_m} ||v|| \leq \lim_{m \to 0}(c_m m^{-1} - 1)||\eta_0|| + c_m n \sup_{u_1 \in U} ||u_1|| + c_m(\lambda - 1) \sup_{u_2 \in Diff} ||u_2|| \tag{39}$$

$$\leq 0 \cdot ||\eta_0|| + 0 \cdot \sup_{u_1 \in U} ||u_1|| + 0 \cdot \sup_{u_2 \in Diff} ||u_2|| \tag{40}$$

$$\leq 0. \tag{41}$$

This limit supremum on $D_m$ tells us that as $m \to 0$, the maximum distance between points in the scaled set $c_m \cdot lpset(m^{-1}\eta_0, n, \lambda - 1)$ and $\eta_0$ gets arbitrarily small. This means there exists some $m_0$ such that for $m < m_0$ the scaled set $c_m \cdot lpset(m^{-1}\eta_0, n, \lambda - 1)$ lies within $\mathcal{B}(\eta_0, \delta_1)$. This scaling mapping can be inverted, and it implies $lpset(m^{-1}\eta_0, n, \lambda - 1)$ is contained within $\frac{1}{c_m}\mathcal{B}(\eta_0, \delta_1)$. Being contained within this scaled ball is precisely what we need to use equation (33) with $\frac{1}{k} = c_m$.

Equation (33) bounds $||\nabla^2 C(\eta)|| \leq H_m = V c_m$ for all $\eta$ in $lpset(m^{-1}\eta_0, n, \lambda - 1)$, which in turn lets us use Lemma 1 to bound our Rényi divergences.

$$D_\lambda(p(\theta|\eta_P)||p(\theta|\eta_Q)) \leq ||\eta_P - \eta_Q||^2 H_m \lambda \tag{42}$$

$$\leq \Delta^2 V c_m \lambda. \tag{43}$$

As we have $c_m \to 0$ as $m \to 0$, we know there must exist $m^*$ such that for $m < m^*$ we have $c_m \leq \frac{\epsilon}{\Delta^2 V \lambda}$. This means the order $\lambda$ Rényi divergences of Algorithm 3 on neighboring data sets is bounded above by $\epsilon$, which gives us the desired result.

$\square$

We have one last theorem to prove, the result claiming the Rényi divergences of order $\lambda$ between $\eta_P$ and its neighbors is convex, which greatly simplifies finding the supremum of these divergences over the convex sets being considered.

**Theorem A.14.** *Let $e(\eta_P, \eta_Q, \lambda) = D_\lambda (p(\theta|\eta_P)||p(\theta|\eta_Q))$.*

*For a fixed $\lambda$ and fixed $\eta_P$, the function $e$ is a convex function over $\eta_Q$.*

*If for any direction $v \in Diff$, the function $g_v(\eta) = v^\intercal \nabla^2 C(\eta)v$ is convex over $\eta$, then for a fixed $\lambda$, the function*

$$f_\lambda(\eta_P) = \sup_{\eta_Q \ r-neighboring \ \eta_P} e(\eta_P, \eta_Q, \lambda) \tag{44}$$

*is convex over $\eta_P$ in the directions spanned by $Diff$.*

PROOF:

First, we can show that for a fixed $\eta_P$ and fixed $\lambda$, the choice of $\eta_Q$ in the supremum must lie on the boundary of possible neighbors. This is derived from showing that $D_\lambda(P||Q)$ is convex over the choice of $\eta_Q$.

Consider once again the expression for our Rényi divergence, expressed now as the function $e(\eta_P, \eta_Q, \lambda)$:

$$e(\eta_P, \eta_Q, \lambda) = D_\lambda(P||Q) = \frac{C(\lambda\eta_P + (1-\lambda)\eta_Q) - \lambda C(\eta_P)}{\lambda - 1} + C(\eta_Q). \tag{45}$$

Let $\nabla_{\eta_Q} e(\eta_P, \eta_Q, \lambda)$ denote the gradient of the divergence with respect to $\eta_Q$.

$$\nabla_{\eta_Q} e(\eta_P, \eta_Q, \lambda) = \nabla C(\eta_Q) + \frac{1-\lambda}{\lambda - 1} \nabla C(\lambda\eta_P + (1-\lambda)\eta_Q) \tag{46}$$

$$= \nabla C(\eta_Q) - \nabla C(\lambda\eta_P + (1-\lambda)\eta_Q). \tag{47}$$

We can further find the Hessian with respect to $\eta_Q$:

$$\nabla^2_{\eta_Q} e(\eta_P, \eta_Q, \lambda) = \nabla^2 C(\eta_Q) - (1-\lambda)\nabla^2 C(\lambda\eta_P + (1-\lambda)\eta_Q). \tag{48}$$

By virtue of being a minimal exponential family, we know $C$ is convex and thus $\nabla^2 C$ is PSD everywhere. Combined with the fact that $\lambda > 1$, this is enough to conclude that $\nabla^2_{\eta_Q} e(\eta_P, \eta_Q, \lambda)$ is also PSD for everywhere with $\lambda > 1$. This means $e(\eta_P, \eta_Q, \lambda)$ is a convex function with respect to $\eta_Q$ for any fixed $\eta_P$ and $\lambda$.

We now wish to characterize the function $f_\lambda(\eta_P)$, which takes a supremum over $\eta_Q \in \eta_P + rDiff$ of $e(\eta_P, \eta_Q, \lambda)$.

$$f_\lambda(\eta_P) = \sup_{\eta_Q r-neighboring \ \eta_P} e(\eta_P, \eta_Q, \lambda) \tag{49}$$

We re-parameterize this supremum in terms of the offset $b = \eta_Q - \eta_P$.

$$f_\lambda(\eta_P) = \sup_{b \in rDiff} e(\eta_P, \eta_P + b, \lambda) \tag{50}$$

Now for any fixed offset $b$, x we can find the expression for the Hessian of $\nabla^2_{\eta_P} e(\eta_P, \eta_P + b, \lambda)$.

$$\nabla^2_{\eta_P} e(\eta_P, \eta_P + b, \lambda) = \nabla^2 C(\eta_P + b) - \frac{\lambda}{\lambda - 1}\nabla^2 C(\eta_P) + \frac{1}{\lambda - 1}\nabla^2 C(\eta_p + (1-\lambda)b) \tag{51}$$

We wish to show this Hessian is PSD, i.e. for any vector $v$ we have $v^\intercal \nabla^2_{\eta_P} e(\eta_P, \eta_P + b, \lambda)v$ is non-negative. We can rewrite this in terms of the function $g_v(\eta)$ introduced in the theorem statement.

(a) $\rho = 1/3$ (high match with $\eta_0$)   (b) $\rho = 1/2$   (c) $\rho = 2/3$ (low match with $\eta_0$)

Figure 1: Utility Comparison for a fixed $\eta_0$ but varying true population parameter

$$v^\mathsf{T} \nabla^2_{\eta_P} e(\eta_P, \eta_P + b, \lambda) v = g_v(\eta_P + b) - \frac{\lambda}{\lambda - 1} g_v(\eta_P) + \frac{1}{\lambda - 1} g_v(\eta_p + (1 - \lambda)b) \tag{52}$$

$$= \frac{\lambda}{\lambda - 1} \left( \frac{\lambda - 1}{\lambda} g_v(\eta_P + b) - g_v(\eta_P) + \frac{1}{\lambda} g_v(\eta_p + (1 - \lambda)b) \right) \tag{53}$$

We know $\frac{\lambda}{\lambda - 1} > 0$ and that $\eta_P$ must lie between $\eta_P + b$ and $\eta_P - (\lambda - 1)b$. Our assumption that $g_v(\eta)$ is convex over $\eta$ for all directions $v$ then lets us use Jensen's inequality to see that the expression (53) must be non-negative.

This lets us conclude that $v^\mathsf{T}(\nabla^2_{\eta_P} e(\eta_P, \eta_P + b, \lambda)v \geq 0$ for all $v$, and thus this Hessian is PSD for any $\eta_P$. This in turn means our divergence $e(\eta_P, \eta_P + b, \lambda)$ is convex over $\eta_P$ assuming a fixed offset $b$.

We return to $f_\lambda(\eta_P)$, and observe that it is a supremum of functions that are convex, and therefore it is convex as well.

□

## A.5 Additional Beta-Bernoulli Experiments

The utility of the prior-based methods (Algorithms 2 and 3) depends on how well the prior matches the observed data. Figure 1 shows several additional situations for the experimental procedure of measuring the log-likelihood of the data.

In each case, the prior $\eta_0 = (6, 18)$ was used, and both $\mathbf{X}$ and $\mathbf{X}_H$ had 100 data points. $\lambda = 15$ was fixed in these additional experiments. The only thing that varies is the true population parameter $\rho$. In (a), $\rho = 1/3$ closely matches the predictions of the prior $\eta_0$. In (b), $\rho = 0.5$, presented as an intermediate case where the prior is misleading. Finally, in (c), $\rho = 2/3$, which is biased in the opposite direction as the prior. In all cases, the proposed methods act conservatively in the face of high privacy, but in (a) this worst case limiting behavior still has high utility. Having a strong informative prior helps these mechanisms. The setting in which the prior is based off of a representative sample of non-private data from the same population as the private data is likely to be beneficial for Algorithms 2 and 3.

One other case is presented in Figure 2, where $\rho = 0.2$ but the prior has been changed to $\eta_0 = (1, 2)$. $\lambda$ is still 15, and the number of data points is still 100. This prior corresponds to the uniform prior, as it assigns equal probability to all estimated data means on $(0, 1)$. It represents an attractive case on a non-informative prior, but also represents a situation in which privacy is difficult. In particular, $\lambda^* = 2$ in this setting. When Algorithm 3 scales up this prior, it becomes concentrated around $\rho = 0.2$, so this setting also corresponds to a case where the true population parameter does not match well with the predictions from the prior.

Figure 2: Utility Experiment for the non-informative uniform prior

## A.6 Application to other exponential families

### A.6.1 Dirichlet-Categorical

The Categorical family is a higher dimension generalization of the Bernoulli family. Instead of just two possible values, (e.g. "failure" or "success", 0 or 1), a categorical variable is allowed to take any of $d$ discrete values. The parameters of a categorical distribution assign a probability to each of the discrete values. These probabilities are constrained to sum to one in order to be a valid distribution, so this family of distribution can be described with only $d - 1$ parameters.

Our propsed method works with this family as well, but the proof is a little more difficult due to the higher dimensions.

Let the space of observations $\mathcal{X} = \{1, 2, \ldots, d\}$. The sufficient statistics of an observation $x$ is a vector of indicator variables, $S(x) = \{\mathbb{I}_1(x), \ldots, \mathbb{I}_{d-1}(x)\}$. Notice that $\mathbb{I}_d(x)$ is not included, since it can be derived from the other coordinates of $S(x)$. Including this last indicator variable would make the family non-minimal where the sufficient statistics satsify the linear relationship $\sum_{i=1}^{d} \mathbb{I}_i = 1$.

The conjugate prior family is the Dirichlet family. Under our construction of the conjugate prior, we want the parameter $\eta$ to satisfy the relationship $\eta_{posterior} = \eta_{prior} + (S(x), 1)$. This means that $\eta$ is $d$ dimensional, and the last coordinate of $\eta$ measures an effective count of observations. Since each coordinate of $S(x)$ is bounded by one, we also have the relationship that for any posterior, $\eta^{(d)} \geq \eta^{(i)}$ for $i \in [d]$.

When $d = 2$, this derivation exactly matches the one from the Beta-Bernoulli system and it is $\Delta$-bounded for $\Delta = 1$.

For $d > 2$, this family is $\Delta$-bounded for $\Delta = \sqrt{2}$. For any two observations, $S(x) - S(y)$ is non-zero in atmost two locations, and each location has a difference of at most 1.

Further, this Dirichlet-Categorical system satisfies the requirements of Theorem A.14. The necessary requirement is that for any direction $v \in Diff$, the function $g_v(\eta) = v^\intercal \nabla^2 C(\eta) v$ is convex over $\eta$. For this system, we have an expression for $C(\eta)$ :

$$C(\eta) = \sum_{k=1}^{d-1} \log\left(\Gamma(\eta^{(k)})\right) + \log\left(\Gamma(\eta^{(d)} - \sum_{i=k}^{d-1} \eta^{(k)})\right) - \log\left(\Gamma(\eta^{(d)})\right) \tag{54}$$

This value is merely the sum of the log-Gamma function applied to the count of observations at each value, minus the log-Gamma function applied to the total count of observations. The expression $\eta^{(d)} - \sum_{i=1}^{d-1} \eta^{(i)}$ evaluates to the count of observations located at the implicit $d^{th}$ value, since $\eta^{(d)}$ carries the total count of observations seen.

With this expression, we can calculate the gradient and Hessian. The digamma function $\psi_0$ is the derivative of the log-Gamma function $\log\left(\Gamma(\cdot)\right)$, and the trigamma function $\psi_1$ is the derivative of the digamma function.

$$\nabla C(\eta)^{(i)} = \begin{cases} \psi_0\left(\eta^{(i)}\right) - \psi_0\left(\eta^{(d)} - \sum_{k=1}^{d-1}\eta^{(k)}\right) & i \neq d \\ \psi_0\left(\eta^{(d)} - \sum_{k=1}^{d-1}\eta^{(k)}\right) - \psi_0\left(\eta^{(d)}\right) & i = d \end{cases} \tag{55}$$

$$\nabla^2 C(\eta)^{(i,j)} = \begin{cases} \psi_1\left(\eta^{(i)}\right) + \psi_1\left(\eta^{(d)} - \sum_{k=1}^{d-1}\eta^{(k)}\right) & i = j \neq d \\ \psi_1\left(\eta^{(d)} - \sum_{k=1}^{d-1}\eta^{(k)}\right) & i \neq j, i \neq d, j \neq d \\ -\psi_1\left(\eta^{(d)} - \sum_{k=1}^{d-1}\eta^{(k)}\right) & i \neq j, i = d, j \neq d \\ -\psi_1\left(\eta^{(d)} - \sum_{k=1}^{d-1}\eta^{(i)}\right) & i \neq j, i \neq d, j = d \\ \psi_1\left(\eta^{(d)} - \sum_{k=1}^{d-1}\eta^{(k)}\right) - \psi_1(\eta^{(d)}) & i = j = d \end{cases} \tag{56}$$

When $v \in Diff$, the last coordinate of $v$ is zero since changing one observation does not change the total count of observations. This means the expression $g_v(\eta) = v^{\mathsf{T}}\nabla^2 C(\eta)v$ can ignore the last coordinate of $v$, as well as the last row and column of $\nabla^2 C(\eta)$. This means we are only concerned with the entries matching the first two cases of equation (56). Let $\tilde{v}$ denote the vector formed by the first $d-1$ coordinates of $v$.

A careful examination the matrix $M$ equal to the top $d-1$ rows and and leftmost $d-1$ columns of $\nabla^2 C(\eta)$ reveals that $M$ decomposes as

$$M = \psi_1\left(\eta^{(d)} - \sum_{k=1}^{d-1}\eta^{(k)}\right)[\mathbf{1}] + diag\left(\psi_1(\eta^{(1)}), \ldots, \psi_1(\eta^{(d-1)})\right) \tag{57}$$

where $[\mathbf{1}]$ is the $d-1$ by $d-1$ matrix where all entries are 1, and $diag$ constructs a diagonal matrix from the given values. This means for all $v \in Diff$, we have the following expression:

$$g_v(\eta) = v^{\mathsf{T}}\nabla^2 C(\eta)v \tag{58}$$
$$= \tilde{v}^{\mathsf{T}}M\tilde{v} \tag{59}$$
$$= \tilde{v}^{\mathsf{T}}\left(\psi_1(\eta^{(d)} - \sum_{k=1}^{d-1}\eta^{(k)})[\mathbf{1}] + diag(\psi_1(\eta^{(1)}, \ldots, \eta^{(d-1)}))\right)\tilde{v} \tag{60}$$
$$= \psi_1(\eta^{(d)} - \sum_{k=1}^{d-1}\eta^{(k)})\left(\tilde{v}^{\mathsf{T}}[\mathbf{1}]\tilde{v}\right) + \sum_{i=1}^{d-1}\psi_1(\eta^{(i)})(\tilde{v}^{(i)})^2 \tag{61}$$
$$\tag{62}$$

With the fact that $[\mathbf{1}]$ is PSD and that $(\tilde{v}^{(i)})^2$ is always positive, the above calculations show that $g_v(\eta)$ is the sum of many applications of the digamma function $\psi_1$. Each of these applications has a positive coefficient, and the function $\psi_1$ is convex. This concludes the proof that $g_v(\eta)$ is convex over $\eta$ for any $v \in Diff$. (When $d = 2$, this expression for $g_v$ in fact matches the one derived from the Beta-Bernoulli system.)

This means that the expression for the worst-case Rényi divergence between neighboring posterior parameters is convex, and so the maximum must be located at the boundaries. In this case, the $pset$ is a shifted simplex, so the maximum must occur at one of the vertices.

The potential pairs of posterior parameters that must be checked in order to evaluate the RDP guarantee grows as $O(d^3)$.

### A.6.2 Gaussian-Gaussian and non-$\Delta$-bounded families

Another interesting setting is estimating the mean of a Gaussian variable when the variance is known. In this case, the conjugate prior is also a Gaussian distribution.

This system satisfies the $v^\intercal \nabla^2 C(\eta) v$ convexity requirement for Theorem A.14, since the variance $\nabla^2 C(\eta)$ is constant when the final coordinate (the total count of observations) is fixed. Thus for any $v \in Diff$, the function $g_v(\eta)$ is constant and therefore convex.

However, this setting does not satisfy the $\Delta$-bounded assumption. The observations can be arbitrarily large, and changing a single observation can therefore lead to arbitrarily large changes to posterior parameters and thus also arbitrarily large Rényi divergences between neighboring data sets.

The exponential family mathematics behind our results did not directly depend on the $\Delta$-boundedness assumption. Instead, this bound was used only to bound the $pset$ of possible posterior parameters in order to bound the distance $||\eta_P - \eta_Q||$ when considering neighboring data sets. This bounded $pset$ then ensured our privacy guarantee was finite.

For any given data set $\mathbf{X}$, we can bound the Rényi divergence between the posterior from $\mathbf{X}$ and the posterior from any other data set $\mathbf{X}'$ satisfying $S(\mathbf{X}) - S(\mathbf{X}') \leq \Delta$. This is true even when the exponential family is not $\Delta$-bounded.

This permits two different approaches: we can relax the RDP framework further, protecting only data sets and a select bounded range of neighboring data sets rather than all the neighbors, or we can include a data preprocessing step that projects the observations onto a set with bounded sufficient statistics. The latter approach permits the use of the RDP framework without introducing further relaxations.

For example, we could replace the observations $\mathbf{X}$ with $\tilde{\mathbf{X}} = f(X)$ where the following function $f$ was applied to each observation $x$ in $\mathbf{X}$:

$$f(x) = \begin{cases} -\Delta & x \leq -\Delta \\ x & -\Delta < x < \Delta \\ \Delta & \Delta \leq x \end{cases} \tag{63}$$

Although the statistical model still believes arbitrarily large observations are possible, the preprocessing projection step allows us to bound $||\eta_P - \eta_Q|| \leq \Delta$ where $\eta_P$ is the posterior for $f(\mathbf{X})$ and $\eta_Q$ is the posterior for $f(\mathbf{X}')$ with any neighboring set of observations $\mathbf{X}'$.

This comes with the caveat that our model no longer matches reality, since it is unaware of the distortions introduced by our preprocessing step. We leads to a potential degradation of utility for the mechanism output, but our privacy guarantees will hold. If the data altered by $f$ is sufficiently rare, these distortions should be minimal.

### A.7 Proofs in Section 4

### A.7.1 GLMs Privacy Proof

In this section we prove Theorem 16 and 17. Here we state and prove a more general version of the theorems. Consider any problem with likelihood in the form

$$p(D|w) = \exp\left(-\sum_{i=1}^n \ell(w, x_i, y_i)\right)$$

and posterior of the following form

$$p(w|D) = \frac{\exp\left(-\sum_{i=1}^n \rho\ell(w, x_i, y_i)\right) p(w)}{\int_{\mathbb{R}^d} \exp\left(-\sum_{i=1}^n \rho\ell(w', x_i, y_i)\right) p(w')dw'}, \tag{64}$$

where in the case of logistic regression, $\ell$ is the logistic loss function.

Then we have the following lemma.

**Lemma 1.** *Suppose $\ell(\cdot, x, y)$ is L-Lipschitz and convex, and $-\log p(w)$ is twice differentiable and $m$-strongly convex. Posterior sampling from (64) satisfies $(\lambda, \frac{2\rho^2 L^2}{m}\lambda)$-RDP for all $\lambda \geq 1$.*

*Proof.* (of Lemma 1) The proof follows from the same idea as in the proof of Theorem 7 of (**?**). The basic idea is that the posterior distribution $p(\cdot|D)$ satisfies Logarithmic Sobolev inequality (LSI), which implies sub-Gaussian concentration on $\log \frac{p(w|D)}{p(w|D')}$; and sub-Gaussian concentration implies RDP.

Before the proof, we define LSI and introduce the relation between sub-Gaussian concentration and LSI.

**Definition A.15.** *A distribution $\mu$ satisfies the Log-Sobolev Inquality (LSI) with constant $C$ if for any integrable function $f$,*

$$\mathbb{E}_\mu \left[ f^2 \log f^2 \right] - \mathbb{E}_\mu \left[ f^2 \right] \log \mathbb{E}_\mu \left[ f^2 \right] \leq 2C\mathbb{E}_\mu \left[ \|\nabla f\|^2 \right].$$

**Theorem A.16.** *(Herbst's Theorem) If $\mu$ satisfies LSI with constant $C$. Then for every $L$-Lipschitz function $f$, for any $\lambda$, $\mathbb{E}\left[ exp\left( \lambda(f - \mathbb{E}_\mu\left[ f \right]) \right) \right] \leq exp\left( C\lambda^2 L^2/2 \right).$*

**Lemma 2.** *Let $U : \mathbb{R}^d \to \mathbb{R}$ be a twice differential, $m$-strongly convex and integrable function. Let $\mu$ be a probability measure on $\mathbb{R}^d$ whose density is proportional to $exp(-U)$. Then $\mu$ satisfies LSI with constant $C = 1/m$.*

Now we prove RDP bound of posterior sampling from (64).

Firstly, notice that negative of log of the prior, $-\log p(w)$, is twice differentiable, $m$-strongly convex and integrable. And therefore negative of log of the posterior, $\rho \sum_{i=1}^n \ell(w, x_i, y_i) - \log p(w)$ is $m$-strongly convex. According to Lemma 2, distribution $p(w|D)$ satisfies LSI with constant $1/m$.

Then, set $f$ in Theorem A.16 as $f(D, D', w) = \log \frac{p(w|D)}{p(w|D')}$. Since the $\ell(\cdot, x, y)$ is $L$-Lipschitz, we know that $f(D, D', w)$ is $2\rho L$-Lipschitz. According to Theorem A.16, for any $\lambda \in \mathbb{R}$,

$$\mathbb{E}_{w\sim p(w|D)} \left[ \exp\left( \lambda \left( \log \frac{p(w|D)}{p(w|D')} - D_{\text{KL}}\left( p(w|D)\|p(w|D') \right) \right) \right) \right] \leq e^{2\lambda^2 \rho^2 L^2/m}.$$

Let $a = 2\rho^2 L^2/m$. Equivalently, then for any $\lambda \in \mathbb{R}$,

$$\mathbb{E}_{w\sim p(w|D)} \left[ \exp\left( \lambda \log \frac{p(w|D)}{p(w|D')} \right) \right] \leq \exp\left( a\lambda^2 + \lambda D_{\text{KL}}\left( p(w|D)\|p(w|D') \right) \right).$$

And setting $\lambda$ to $\lambda - 1$, we have

$$\mathbb{E}_{w\sim p(w|D)} \left[ \exp\left( (\lambda-1) \log \frac{p(w|D)}{p(w|D')} \right) \right] \leq \exp\left( a(\lambda-1)^2 + (\lambda-1)D_{\text{KL}}\left( p(w|D)\|p(w|D') \right) \right)$$
$$\leq \exp\left( (\lambda-1)\left( a\lambda + D_{\text{KL}}\left( p(w|D)\|p(w|D') \right) - a \right) \right).$$

If $\lambda \geq 1$, the expectation is upper bounded by

$$\exp\left( (\lambda-1)\left( a\lambda + \max_{d(D,D')=1} D_{\text{KL}}\left( p(w|D)\|p(w|D') \right) - a \right) \right).$$

According to the definition of zCDP in (**?**), this implies zCDP with

$$\rho = \frac{2\rho^2 L^2}{m},$$

$$\xi = \max_{d(D,D')=1} D_{\text{KL}}\left( p(w|D)\|p(w|D') \right) - \frac{2\rho^2 L^2}{m},$$

which is equivalent to $(\lambda, \frac{2\rho^2 L^2}{m}\lambda + \max_{d(D,D')=1} D_{\text{KL}}\left( p(w|D)\|p(w|D') \right) - \frac{2\rho^2 L^2}{m})$-RDP for any $\lambda \geq 1$.

Finally, we aim at bounding $D_{\text{KL}}\left( p(w|D)\|p(w|D') \right)$. Let $F(w) = \frac{p(w|D)}{p(w|D')}$. According to the definition of KL-divergence, we have

$$D_{\text{KL}}\left( p(w|D)\|p(w|D') \right) = \mathbb{E}_{p(w|D)}\left[ \log F \right] = \mathbb{E}_{p(w|D')}\left[ F \log F \right] - \mathbb{E}_{p(w|D')}\left[ F \right] \mathbb{E}_{p(w|D')}\left[ \log F \right],$$

which, by setting $f = \sqrt{F}$ in Definition A.15 and having $C = 1/m$, can be upper bounded by

$$D_{\text{KL}}\left(p(w|D)\|p(w|D')\right) \leq \frac{2}{m}\mathbb{E}_{p(w|D')}\left[\|\nabla\sqrt{F}\|_2^2\right]. \tag{65}$$

We have

$$
\begin{aligned}
&\|\nabla \log F\|_2 \\
=&\rho\|\nabla \log p(w|D) - \nabla \log p(w|D')\|_2 \\
=&\rho\|\nabla \log(p(D|w)p(w)) - \nabla \log(p(D'|w)p(w))\|_2 \\
=&\rho\|\nabla \log p(D|w) - \nabla \log p(D'|w)\|_2 \\
\leq&2\rho L,
\end{aligned}
$$

and therefore

$$\|\nabla\sqrt{F}\|_2^2 = \|\nabla\exp\left(\frac{1}{2}\log F\right)\|_2^2 = \|\frac{\sqrt{F}}{2}\nabla \log F\|_2^2 = \frac{F}{4}\|\nabla \log F\|_2^2 \leq \rho^2 L^2 F.$$

So the KL-divergence in (65) is upper bounded by

$$\frac{2\rho^2 L^2}{m}\mathbb{E}_{p(w|D')}\left[F\right] = \frac{2\rho^2 L^2}{m}.$$

Therefore Bayesian logistic regression satisfies $(\lambda, \frac{2\rho^2 L^2}{m}\lambda)$-RDP for any $\lambda$.

For readers familiar with the proof of Theorem 7 in (**?**), the proof here is exactly the same except that the tail bound of sub-Gaussian concentration in Equation 21 and consequently 25 there are replaced by the moment generating function bound. The reason for not using the tail bound to imply moment generating function bound is because that loses constant factor. □

For GLMs, we have

$$\ell(w, x, y) = -\log h(y) + A(w^\top x) - yw^\top x,$$

and thus

$$\nabla_w \ell(w, x, y) = (\mu - y)x = (g^{-1}(w^\top x) - y)x.$$

Then, by the condition in Theorem 16 and 17, $\|\nabla_w \ell(w, x, y)\|_2$ is upper bounded by $Bc$ and $\ell(\cdot, x, y)$ is $Bc$-Lipschitz.

### A.7.2 Logistic Regression Tightness

**Lemma 3.** *For any $d > 1$ and any $n \geq 1$, there exists neighboring datasets $D$ and $D'$, each of size $n$, such that for any $\lambda$, Rényi Divergence for logistic regression with Gaussian prior is of order $\frac{1}{2n\beta}(\lambda - 1)$.*

*Proof.* (of Lemma 3) For convenience, here we assume $\mathcal{Y} = \{-1, 1\}$ instead of $0, 1$, and thus $p(y|w, x)$ can be written as $1/(1 + e^{-yw^\top x})$ for both values of $y$.

Consider any $x \in \mathbb{R}^d$ with $\|x\| = 1$. Let $D = \{(x, y)\}$ and $D' = \{(x', y')\}$, where $x' = x$ and $y' = -y = -1$. Let $p(D|w)$ be the probability of seeing dataset $D$ given classifier $w$, we have

$$\int \frac{p(w|D)^\lambda}{p(w|D')^{\lambda-1}}dw = \int p(w)\frac{p(D|w)^\lambda}{p(D'|w)^{\lambda-1}}dw \times \frac{[\int p(w)p(D'|w)dw]^{\lambda-1}}{[\int p(w)p(D|w)dw]^\lambda}. \tag{66}$$

Let $\sigma^2 = (n\beta)^{-1}$ denote the variance of the Gaussian prior.

Firstly, we prove an equation that will be used later. Let $x_j$ be the $j$-th dimension of $x$ and $w_j$ be the $j$-th dimension of $w$, for any $i$, we have

$$\frac{1}{\sqrt{2\pi\sigma^2}^d} \int_{\mathbb{R}^d} \exp\left(-\frac{\|w\|^2}{2\sigma^2}\right) \exp\left(-iw^\top x\right) dw \tag{67}$$

$$= \prod_{j=1}^{d} \frac{1}{\sqrt{2\pi\sigma^2}} \int_{\mathbb{R}} \exp\left(-\frac{w_j^2}{2\sigma^2}\right) \exp\left(-iw_j x_j\right) dw_j$$

$$= \prod_{j=1}^{d} \frac{1}{\sqrt{2\pi\sigma^2}} \int_{\mathbb{R}} \exp\left(-\frac{(w_j + ix_j\sigma^2)^2}{2\sigma^2}\right) \exp\left(\frac{i^2 x_j^2 \sigma^2}{2}\right) dw_j$$

$$= \prod_{j=1}^{d} \exp\left(\frac{i^2 x_j^2 \sigma^2}{2}\right)$$

$$= \exp\left(\frac{i^2 \sigma^2 \|x\|_2}{2}\right).$$

Now we will consider the two terms in (66) separately for $D$, $D'$ specified above.

For the first term, we have

$$\int_{\mathbb{R}^d} p(w) \frac{p(D|w)^\lambda}{p(D'|w)^{\lambda-1}} dw$$

$$= \frac{1}{\sqrt{2\pi\sigma^2}^d} \int_{\mathbb{R}^d} \exp\left(-\frac{\|w\|^2}{2\sigma^2}\right) \frac{(1+\exp\left(-y'w^\top x'\right))^{\lambda-1}}{(1+\exp\left(-yw^\top x\right))^\lambda} dw$$

$$= \frac{1}{\sqrt{2\pi\sigma^2}^d} \int_{\mathbb{R}^d} \exp\left(-\frac{\|w\|^2}{2\sigma^2}\right) \frac{(1+\exp\left(w^\top x\right))^{\lambda-1}}{(1+\exp\left(-w^\top x\right))^\lambda} dw$$

$$= \frac{1}{\sqrt{2\pi\sigma^2}^d} \int_{\mathbb{R}^d} \exp\left(-\frac{\|w\|^2}{2\sigma^2}\right) \frac{\exp\left((\lambda-1)w^\top x\right)}{1+\exp\left(-w^\top x\right)} dw.$$

Let $S_+$ be any half-space of $\mathbb{R}^d$ and $S_- = \mathbb{R}^d \backslash S_+$. The above equals to

$$\frac{1}{\sqrt{2\pi\sigma^2}^d} \int_{S_+} \exp\left(-\frac{\|w\|^2}{2\sigma^2}\right) \frac{\exp\left(\lambda - 1w^\top x\right)}{1+\exp\left(-w^\top x\right)} dw + \frac{1}{\sqrt{2\pi\sigma^2}^d} \int_{S_-} \exp\left(-\frac{\|w\|^2}{2\sigma^2}\right) \frac{\exp\left(\lambda - 1w^\top x\right)}{1+\exp\left(-w^\top x\right)} dw$$

For any $x$ and any $w \in S_-$, we have $-yw^\top x = y(-w)^\top x$. By changing variable in the second integral from $w$ to $-w$, the above equals to

$$\frac{1}{\sqrt{2\pi\sigma^2}^d} \int_{S_+} \exp\left(-\frac{\|w\|^2}{2\sigma^2}\right) \frac{\exp\left((\lambda-1)w^\top x\right)}{1+\exp\left(-w^\top x\right)} dw + \frac{1}{\sqrt{2\pi\sigma^2}^d} \int_{S_+} \exp\left(-\frac{\|w\|^2}{2\sigma^2}\right) \frac{\exp\left(-(\lambda-1)w^\top x\right)}{1+\exp\left(w^\top x\right)} dw$$

$$= \frac{1}{\sqrt{2\pi\sigma^2}^d} \int_{S_+} \exp\left(-\frac{\|w\|^2}{2\sigma^2}\right) \left(\frac{\exp\left((\lambda-1)w^\top x\right)}{1+\exp\left(-w^\top x\right)} + \frac{\exp\left(-(\lambda-1)w^\top x\right)}{1+\exp\left(w^\top x\right)}\right) dw$$

$$= \frac{1}{\sqrt{2\pi\sigma^2}^d} \int_{S_+} \exp\left(-\frac{\|w\|^2}{2\sigma^2}\right) \sum_{i=-\lambda+1}^{\lambda-1} \exp\left(-iw^\top x\right) (-1)^{i+\lambda-1} dw$$

$$= \sum_{i=-\lambda+1}^{-1} \frac{(-1)^{i+\lambda-1}}{\sqrt{2\pi\sigma^2}^d} \int_{S_+} \exp\left(-\frac{\|w\|^2}{2\sigma^2}\right) \exp\left(-iw^\top x\right) dw + \frac{(-1)^{\lambda-1}}{\sqrt{2\pi\sigma^2}^d} \int_{S_+} \exp\left(-\frac{\|w\|^2}{2\sigma^2}\right) dw$$

$$+ \sum_{i=1}^{\lambda-1} \frac{(-1)^{i+\lambda-1}}{\sqrt{2\pi\sigma^2}^d} \int_{S_+} \exp\left(-\frac{\|w\|^2}{2\sigma^2}\right) \exp\left(-iw^\top x\right) dw.$$

The middle term equals to $(-1)^{\lambda-1}/2$. And changing variable from $w$ to $-w$ in the first term, the above equals to

$$
\begin{aligned}
= & \sum_{i=1}^{\lambda-1} \frac{(-1)^{-i+\lambda-1}}{\sqrt{2\pi\sigma^2}^d} \int_{S_-} \exp\left(-\frac{\|w\|^2}{2\sigma^2}\right) \exp\left(-iw^\top x\right) dw \\
& + \sum_{i=1}^{\lambda-1} \frac{(-1)^{i+\lambda-1}}{\sqrt{2\pi\sigma^2}^d} \int_{S_+} \exp\left(-\frac{\|w\|^2}{2\sigma^2}\right) \exp\left(-iw^\top x\right) dw + \frac{(-1)^{\lambda-1}}{2} \\
= & \sum_{i=1}^{\lambda-1} \frac{(-1)^{-i+\lambda-1}}{\sqrt{2\pi\sigma^2}^d} \int_{\mathbb{R}^d} \exp\left(-\frac{\|w\|^2}{2\sigma^2}\right) \exp\left(-iw^\top x\right) dw + \frac{(-1)^{\lambda-1}}{2}.
\end{aligned}
$$

Using the equation in (67) with the fact that $\|x\|_2 = 1$, the above equals to

$$
\sum_{i=1}^{\lambda-1}(-1)^{-i+\lambda-1}\exp\left(\frac{i^2\sigma^2}{2}\right) + \frac{(-1)^{\lambda-1}}{2} = \sum_{i=0}^{\lambda-1}(-1)^{-i+\lambda-1}\exp\left(\frac{i^2\sigma^2}{2}\right).
$$

As for the second term in (66), we have

$$
\int p(w)p(D|w)dw = \frac{1}{\sqrt{2\pi\sigma^2}^d} \int_{\mathbb{R}^d} \exp\left(-\frac{\|w\|^2}{2\sigma^2}\right) \frac{1}{1+\exp\left(-yw^\top x\right)}dw.
$$

Let $S_+$ be any half-space of $\mathbb{R}^d$ and $S_- = \mathbb{R}^d\backslash S_+$. The above equals to

$$
\frac{1}{\sqrt{2\pi\sigma^2}^d} \int_{S_+} \exp\left(-\frac{\|w\|^2}{2\sigma^2}\right) \frac{1}{1+\exp\left(-yw^\top x\right)}dw + \frac{1}{\sqrt{2\pi\sigma^2}^d} \int_{S_-} \exp\left(-\frac{\|w\|^2}{2\sigma^2}\right) \frac{1}{1+\exp\left(-yw^\top x\right)}dw.
$$

For any $x$ and any $w \in S_-$, we have $-yw^\top x = y(-w)^\top x$. By changing variable in the second integral from $w$ to $-w$, the above equals to

$$
\begin{aligned}
& \frac{1}{\sqrt{2\pi\sigma^2}^d} \int_{S_+} \exp\left(-\frac{\|w\|^2}{2\sigma^2}\right) \frac{1}{1+\exp\left(-yw^\top x\right)}dw + \frac{1}{\sqrt{2\pi\sigma^2}^d} \int_{S_+} \exp\left(-\frac{\|w\|^2}{2\sigma^2}\right) \frac{1}{1+\exp\left(yw^\top x\right)}dw \\
= & \frac{1}{\sqrt{2\pi\sigma^2}^d} \int_{S_+} \exp\left(-\frac{\|w\|^2}{2\sigma^2}\right) \left(\frac{1}{1+\exp\left(-yw^\top x\right)} + \frac{1}{1+\exp\left(yw^\top x\right)}\right) dw \\
= & \frac{1}{\sqrt{2\pi\sigma^2}^d} \int_{S_+} \exp\left(-\frac{\|w\|^2}{2\sigma^2}\right) dw \\
= & \frac{1}{2}.
\end{aligned}
$$

The above results holds for any $D$, which implies that the second term of (66) equals to 2.

Therefore, (66) equals to

$$
2\sum_{i=0}^{\lambda-1}(-1)^{-i+\lambda-1}\exp\left(\frac{i^2\sigma^2}{2}\right).
$$

As $(\lambda - 1) \to \infty$, this formula is of order

$$
\mathcal{O}\left(\exp\left(\frac{(\lambda-1)^2\sigma^2}{2}\right)\right).
$$

$\square$

## A.8 Additional Experiments for GLMs

In this section, we present more experimental results on the same datasets with slightly different privacy requirement.

Figure 3: Test error vs. privacy parameter $\epsilon$. $\lambda = 1$ for upper row; $\lambda = 10$ for lower row.

Firstly we present the test error of all algorithms under the three dataset at $\lambda = 1$ and $10$ in Figure 3. The same pattern as in $\lambda = 100$ can be observed – both of our proposed algorithms achieve better utility than OPS, and the diffused algorithm is always better than the concentrate algorithm. We can also see the degradation in utility as $\lambda$ increase.

We next show the negative log-likelihood at $\lambda \in \{1, 10, 100\}$ in Figure 4. We can see the same trend as that in test error. Both of our proposed algorithms achieves smaller negative log-likelihood, and the diffused algorithm achieves lower negative log-likelihood than the concentrate algorithm.

Figure 4: Negative log-likelihood vs. privacy parameter $\epsilon$. $\lambda = 1, 10, 100$ from upper to lower row. y-axis plotted in log scale.