[Reviews · NeurIPS 2017]

Reviewer 1



This paper proposes two tunable posterior sampling mechanisms for exponential families for Renyi Differential Privacy. (Although not clear from the exposition, RDP is a known variant of DP) Both of them, just like recent approaches, rely on manipulating the statistics. The first, Diffused Posterior, reduces the weight of the observed data (in a manner similar to the epsilon parameter in [13]). The second one increases the weight of the prior instead (analogously to the concentration constant in [3]). In fact, it would probably be nice if the authors could expand a bit on the similarities between these ideas. The paper provides also a nice pair of examples, where they demonstrate the practicality of these mechanisms. No obvious typos apart from a repetition of refs 9, 10.

Reviewer 2



This paper analyzes the privacy cost of posterior sampling for exponential family posteriors and generalized linear models by using the recently proposed privacy definition of Renyi-DP. Specifically, they applied the privacy analyses to two domains and presented experimental results for Beta-Bernoulli sampling and Bayesian Logistic Regression. In my opinion, the privacy analyses that the paper considers can be interesting and valuable for the community. But, the paper does not motivate enough the necessity of the usage of Renyi-DP. It is a relaxed notion of pure DP like CDP and z-CDP but the readers cannot learn from the paper that when they can use it or why should they use it. In terms of technical correctness of the the paper, the algorithm presented relies on RDP and standard DP definitions, I believe the analyses are correct. But, the problem is the authors did not cite the definitions which are already published in the literature (e.g. Definition 1-2-3-4-…). So, it is hard for me (and I suppose for the other readers) to distinguish which of the definitions/theorems are proposed by the authors and which of them are already published. Furthermore, the search process to find an “appropriate” value of r and m is not clear. In the experiments section they searched over six values but we do not know whether all of them are valid or not. What do you mean by “appropriate”? Besides, I prefer to see the connection/correlation between the values r/m and accuracy (their intuitive interpretation would be immensely helpful). Maybe I miss it, but the paper does not provide any formal utility guarantee, which is something interesting (at least we can see the privacy/accuracy connection empirically). Finally, in Figure 2c and 2d and in the real data experiments, there is no baseline such as the non-private case. OPS is not a proper baseline because we cannot compare pure-DP and RDP (maybe you can formalize the connection somewhere in methodology and then we can compare). So, it will be nice to show what is a good result for these experiments when privacy is not a concern.